# Comparing a Query Compound with Drug Target Classes Using 3D-Chemical Similarity

**DOI:** 10.3390/ijms21124208

**Published:** 2020-06-12

**Authors:** Sang-Hyeok Lee, Sangjin Ahn, Mi-hyun Kim

**Affiliations:** 1Gachon Institute of Pharmaceutical Science and Department of Pharmacy, College of Pharmacy, Gachon University, Yeonsu-gu, Incheon 21936, Korea; prizeh83@gmail.com; 2Innovation Center for Industrial Mathematics, National Institute for Mathematical Science, Yeongtong-gu, Suwon 16229, Korea; 3Department of Financial Engineering, College of Business, Ajou University, Suwon 16499, Korea; asj92@ajou.ac.kr

**Keywords:** Kullback–Leibler (K–L) divergence, chemocentric similarity, Jaccard–Tanimoto coefficient, Gaussian mixture model (GMM), expectation-maximization (EM) algorithm, maximum likelihood (ML) estimation, machine learning

## Abstract

3D similarity is useful in predicting the profiles of unprecedented molecular frameworks that are 2D dissimilar to known compounds. When comparing pairs of compounds, 3D similarity of the pairs depends on conformational sampling, the alignment method, the chosen descriptors, and the similarity coefficients. In addition to these four factors, 3D chemocentric target prediction of an unknown compound requires compound–target associations, which replace compound-to-compound comparisons with compound-to-target comparisons. In this study, quantitative comparison of query compounds to target classes (one-to-group) was achieved via two types of 3D similarity distributions for the respective target class with parameter optimization for the fitting models: (1) maximum likelihood (ML) estimation of queries, and (2) the Gaussian mixture model (GMM) of target classes. While Jaccard–Tanimoto similarity of query-to-ligand pairs with 3D structures (sampled multi-conformers) can be transformed into query distribution using ML estimation, the ligand pair similarity within each target class can be transformed into a representative distribution of a target class through GMM, which is hyperparameterized via the expectation–maximization (EM) algorithm. To quantify the discriminativeness of a query ligand against target classes, the Kullback–Leibler (K–L) divergence of each query was calculated and compared between targets. 3D similarity-based K–L divergence together with the probability and the feasibility index, (F_m_), showed discriminative power with regard to some query–class associations. The K–L divergence of 3D similarity distributions can be an additional method for (1) the rank of the 3D similarity score or (2) the *p*-value of one 3D similarity distribution to predict the target of unprecedented drug scaffolds.

## 1. Introduction

An unpresented molecular framework such as that in Figure 1a can be investigated in drug space. In early stages of drug discovery, three-dimensional (3D) similarity between chemicals has been used to find desirable ligands of a chosen therapeutic target in virtual screening (VS; Figure 1b) [1,2]. To our knowledge, chemical similarity is a coarse predictor for filtering out less promising chemicals rather than selecting the most desirable compound. Chemical similarity has also contributed to target screening (in other words, retro-VS) under the chemocentric assumption in Figure 1c. Chemocentric assumption means if two similar molecules are likely to possess similar properties, they can share biological targets or may show similar pharmacological profiles [3,4]. Remarkably, Jain’s group conducted on-target and off-target prediction through the comparison of two-dimensional (2D) and 3D chemical similarity [5]. Based on this comparison, while dual 2D and 3D similarity-based predictions showed superiority for either 2D or 3D predictions, 3D predictions did not show dramatic improvement over 2D predictions. In addition, the increase of data points, according to the conformer sampling sizes, makes the computing cost of 3D features increase more rapidly than 2D features. However, despite it being less cost-effective, 3D similarity is the best feature for in silico target screening of unprecedented drug scaffolds and new drug-like molecular frameworks [6] because (1) novel, unprecedented drug scaffolds have very low 2D similarity to known bioactive molecules [7,8,9], (2) novel pharmacological profiles of drugs are more frequently found using 3D similar off-target predictions [5], and (3) realistic drug properties can be generated from their factual and flexible 3D structures [10,11,12].

The internalization of Michelangelo Buonarroti’s quote, “Every block of stone (chemical) has a statue (utility) inside it, and it is the task of the sculptor (chemist) to discover it”, inspired this research for the ‘chemistry-oriented synthesis’ of an unprecedented drug scaffold [7,8,9] and the chemocentric target profiling of this scaffold [7]. For this purpose, we have intensively studied the 3D similarity of unprecedented drug scaffolds (the query compounds) with known molecular frameworks (the reference compounds). When comparing query and reference compound pairs, 3D similarity of the pairs depends on (1) conformational sampling of the compounds, (2) the alignment method, (3) the chosen descriptors, and (4) the distance coefficients (e.g., Jaccard–Tanimoto). In addition to the four factors of 3D VS, retro-VS of unprecedented drug scaffolds (query compounds) requires compound–target associations (target class information), as shown in Figure 1. These associations are the source of the substantial difference between VS and retro-VS in problem-solving in data science, specifically, (1) one-to-one comparison for VS, as shown in Figure 1b; (2) one-to-group (class) comparison for retro-VS, as shown in Figure 1c; and (3) group-to-group comparison for typical parametric statistics such as ANOVA and *t*-test. When we calculated the similarity of compound pairs in retro-VS, the hope was to ultimately identify the primary target of the query through calculated chemical similarity rather than finding the most similar compound to the query structure. To achieve this, one-to-group comparison must be essentially quantified. To our knowledge, such measurements have not been properly reported in cheminformatics. Notably, 2D similarity distributions with target annotation have been reported using statistical fitting models such as Shoichet’s group [3], Bajorath’s group [13], and Nasr’s group [14]. However, even though the number of studies using 3D similarity is enormous with review articles by Zhang et al. [15] and Shin et al. [16], 3D similarity distribution is rarely mentioned in the literature. Other than the distribution, network analysis (edge: similarity, node: chemical) such as that by Torres et al. [17] or the machine-learning algorithm-based classifiers have also been used [11,18]. Most classifiers do not only use chemical similarity, but also use other descriptors together [18]. Although several studies have treated 3D similarity distribution such as Jain’s group [5], Medina-Franco’s group [19], and Pérez-Nueno’s group [20], the distribution comprised every compound instead of compounds grouped by target [5,19]. In addition, it was either visualized without a fitting model [19] or its statistical model was chosen without parameter optimization [5]. Exceptionally, although Pérez-Nueno’s group reported Gaussian distribution using 3D similarity, the study assumed Gaussian distribution with only one centroid and fitting parameter was also not optimized, despite the small number of ligands [20].

In this study, we quantitatively compared a query compound with a target class (one-to-group) using two types of similarity distributions, namely, maximum likelihood (ML) estimation of queries and a Gaussian mixture model (GMM) of target classes (Figure 1d). As raw data of this study, the Jaccard–Tanimoto similarity coefficients were calculated for (1) query-to-ligand pairs (e.g., the left second row of the Figure 1d) and (2) ligand pairs within each target class (e.g., the left first row of Figure 1d). The query-to-ligand similarity was transformed into query distribution via ML estimation, and the ligand pair similarity was also transformed into a representative distribution of a target class using GMM. The difference between two distributions was quantified by Kullback–Leibler (K–L) divergence, which represented the quantitative comparison between a query and a target class. In order to evaluate whether the K–L divergence accurately achieved one-to-group comparison, a query chosen from a group of known ligands for a target was tested to observe discrimination between the original target and other targets. In sequence, the target profiles of an unprecedented drug scaffold was explained by K–L divergence.

## 2. Theoretical Background

**Kullback–Leibler divergence:** K–L divergence measures the difference between two statistical or probabilistic distributions. In particular, K–L divergence is employed in various machine learning and deep learning algorithms for statistical inference [21,22]. Since K–L divergence implies relative entropy, which is an important concept in understanding statistical phenomena, it applies to statistical physics, chemistry, and social science.

Let us define two probability spaces, Ω,F,P and Ω,F,Q, where Ω is the sample space, F is σ–algebra, and P and Q are probability distributions. Then, to define Kullback–Leibler divergence, a unique measurable function is devised, dQdP:Ω→ℝ+, known as the Radon–Nykodym derivative, so that
(1)QE=∫EdQdPdP


For any measurable set, E∈Ω [22] when using the measurable function $\frac{dQ}{dP}$. The Kullback–Leibler divergence, D(P∥Q), is defined as either
(2)$$D\left(P∥Q\right):=\int_{\Omega }-ln\left(\frac{dP}{dQ}\right)dP$$
or
(3)$$D\left(P∥Q\right):=\int_{-\infty }^{\infty }ln\left(\frac{p\left(x\right)}{q\left(x\right)}\right)p\left(x\right)dx,$$
where the probability density functions *p*(*x*) and *q*(*x*) are defined as
(4)$$P\left(x\right):=\int_{-\infty }^{x}p\left(x\right)dx\;\mathrm{and}\;Q\left(x\right):=\int_{-\infty }^{x}q\left(x\right)dx$$


The Kullback–Leibler divergence represents the information for comparing *P*(*x*) and *Q*(*x*) distributions [23]. Hence, the implication of Kullback–Leibler divergence depends on the definitions of *P*(*x*) and *Q*(*x*). For example,
Model Inference: If *P*(*x*) represents the testing distribution based on the model, and *Q*(*x*) represents the distribution from the raw data, the difference is the error between the model and reality [24];Informatics: If *P*(*x*) and *Q*(*x*) represent information extracted from two objectives, the divergence is a measurement for the discrimination between two objectives [13,25];Bayesian Statistics: If *P*(*x*) represents a prior distribution and *Q*(*x*) represents a posterior distribution, the divergence represents the information gained through updating [26,27].


In sequence, let us consider a special example. Assume the probability distributions *P*(*x*) and *Q*(*x*) replace the Gaussian distributions $G\left(x;m_{i},\sigma_{i}\right)$ and $G\left(x;m_{j},\sigma_{j}\right)$, where
(5)$$G\left(x;m_{i},\sigma_{i}\right):=\int_{-\infty }^{x}g\left(s;m_{i},\sigma_{i}\right)ds\;\mathrm{and}\;G\left(x;m_{j},\sigma_{j}\right):=\int_{-\infty }^{x}g\left(s;m_{j},\sigma_{j}\right)ds$$


Using Equations (3) and (5), the Kullback–Leibler divergence between the two Gaussian distributions $G\left(x;m_{i},\sigma_{i}\right)$ and $G\left(x;m_{j},\sigma_{j}\right)$ in Equation (5) are as follows:
(6)$$D\left(G\left(x;m_{i},\sigma_{i}\right)∥G\left(x;m_{j},\sigma_{j}\right)\right)=ln\left(\frac{\sigma_{j}}{\sigma_{i}}\right)+\frac{{\left(\sigma_{i}\right)}^{2}+{\left(m_{i}-m_{j}\right)}^{2}}{2{\left(\sigma_{j}\right)}^{2}}-\frac{1}{2}$$


This Kullback–Leibler divergence between the univariate normal distributions (Equation (6)) therefore extends to multivariate distributions [28].

**Gaussian mixture model:** The mixture models are methods that analyze compositional data. With Φ representing a probabilistic density generated from the unknown compositional data, p representing a well-known probability density, and **x** representing a random vector, the functional operator, $\Xi \left(\Phi \left(x\right)|p,K\right)$, is defined as
(7)$$\Xi \left(\Phi \left(x\right)|p,\omega ,\lambda ,K\right):=\sum_{k=1}^{K}\omega_{k}p\left(x:\lambda_{k}\right)$$
where for k = 1, 2, …, K, $\omega_{k}$, $\lambda_{k}$ are the weights and vectors of the hyperparameters and $p_{i}$ is the $i_{th}$ component, which is independently and identically distributed (iid) [29]. In this work, GMM was adopted to obtain a representative distribution [30]. Notably, GMM is a model that describes non-Gaussian distributions as well as Gaussian distributions [31]. The probability density $p\left(x:\lambda_{k}\right)$ represents the Gaussian density function $g\left(x;m_{k},\sigma_{k}\right)$ in Equation (5). In the Gaussian mixture model, estimations of the weight ($\omega_{k}$), the mean ($m_{k}$), and the standard deviation ($\sigma_{k}$) are essential. Herein, the two methods (i.e., the EM algorithm [32] and ML estimation [33]) were chosen to estimate the hyperparameters from sparse and incomplete data. The EM algorithm for GMM consists of an initial guess for the GMM parameters and iterative calculation (E-step)–parameter determination (M-step). The iterative steps continue until the set of hyperparameters, θ, are less than positive, and infinitesimal number, ϵ, as shown in the ccccccmathematical elucidation (Appendix A [34]. For convenience, when applying the ML estimation, $\Phi \left(x\right)$ is transformed into the mixture model and $\Xi \left(\Phi \left(x\right)|p,\omega ,\lambda ,K\right)$ is replaced by $\Xi_{EM}\left(\Phi \left(x\right)|p,\omega ,\lambda ,K\right)$.

## 3. Results and Discussion

In this study, a quantitative method was developed to describe discriminative information for target prediction of a query compound only from chemical similarity and known compound–target association information. For this purpose, 3D similarity distributions were acquired from a 3D similarity matrix occupied by Jaccard–Tanimoto coefficients [35] regarding (1) query-to-ligand pairs and (2) ligand pairs within each target class. The Jaccard–Tanimoto coefficients were calculated from two types of features, molecular shape and pharmacophore features, using the Openeye Toolkit. Query compounds and target classes were compared and quantified according to the following process:
**Step 1.** EM algorithm-based GMM allowed to obtain a representative distribution (Q-distribution) for a target class, following either Gaussian or non-Gaussian distribution;**Step 2.** A query-to-ligand similarity distribution was fitted onto a Gaussian distribution using ML estimation;**Step 3.** K–L divergence between the two distributions from Step 1 and Step 2 allowed target predictions of the query compound. Greater deviation of K–L divergence values between target classes indicated that the query compound was a more representative ligand of a class than other query compounds. In addition, the probability, ℙνlm=i, derived from the K–L divergence values and the feasibility index, Fm, allowed for quantification of discrimination between the target classes.


**Dataset:** In order to select example target classes for this study, an unprecedented scaffold with structural novelty and its targets were focused. Among our previous studies, bis-*N*,*N*-dimethylaminophenylamino tetrahydropyran (BNDS-A), which was the most potent to regulate in vitro inflammation (IC_50_ of nitric oxide production = 12 μM), was chosen for this quantitative method (Figure 1a). The association of two targets with BNDS-A, estrogen receptor alpha (ESR), and vitamin D receptor (VDR) was proven by the stepwise approach consisting of (1) 2D similarity search, (2) multiplication of 3D similarity coefficients of every ligand within each target, P(Tc)/C(hits), (3) self/cross-similarity, and (4) western blot analysis in our previous work [7]. However, despite low predicted probability, capthesin D (CTSD) and cyclooxygenase-2 (COX2) could also be regulated by BNDS-A in the same study. Neither the most similar compound to BNDS-A (one-to-one comparison) nor ANOVA test between target pairs (group-to-group comparison) could suggest the primary target of BNDS-A. Therefore, to quantitatively compare them with BNDS-A, the four targets, ESR, VDR, COX2, and CTSD, were selected. In addition, an additional four targets, HIV-1 protease (HIV1), heat shock protein 90 (HSP90), transient receptor potential cation channel subfamily V4 (TRPV4), DNA topoisomerase I (TOP1), were randomly selected from the target prediction literature [36] to evaluate our methodology. For convenience, simple numbers denoted the target classes, in other words,
(8)$$\left\{\begin{array}{c} Estrogen\;receptor\;alpha\to 1,\\ Vitamin\;D\;receptor\to 2,\\ Cyclooxygenase-2\to 3,\\ Cathepsin\;D\to 4. \end{array}\right.$$


Either *m* or *n* were called the class number, which was an integer between 1 and 4, as in Equation (8), and CLm and CLn∈ℝN represent vectors whose elements are the Tanimoto coefficients of query compounds in the *m*th class. TM:ℝ2N→ℝN×ℝN was defined as the Tanimoto matrix operator, so
(9)TMClm,Clnij:=Tc<ei·Clm>,<ej,Cln>
where Tci,j is a scalar operator between the *i*th and *j*th queries to calculate the Tanimoto coefficient and ei and ej are unit vectors for the *i*-axis and *j*-axis, where <, > is the inner product.

**Representative distributions *Q* for target classes:** The representative distributions corresponding to each target class using GMM of ligand pair similarity were obtained. First, using the similarity matrix TMClm,Clnij in Equation (9), where m = n, the following univariate probability densities, Φnxk, were defined by
(10)Φnxiδx:=ℙxk≤X=TMClm,Clnij≤xk+1,
where ℙ is the probability measure; *x* is the Tanimoto–Jaccard coefficients; 0=x0 and the range of *x* is [0, 2]; and xk+1=xk+δx. Therefore, the probability densities, Φnx, satisfy the following equation:
(11)∑i=0999Φnxiδx=1


Second, to extract representative distributions from $\Phi_{n}\left(x\right)$, the Gaussian mixture model was utilized, in which probability densities, $\Phi_{n}\left(x\right)$, are expressed as approximated from $\Xi_{EM}\left(\Phi_{n}\left(x\right)|G,\omega ,\mu ,\sigma ,K\right)$, which is the weighted sum of K univariate Gaussian distributions. That is,
(12)$$\Xi_{EM}\left(\Phi_{n}\left(x\right)|g,\omega ,\mu ,\sigma ,K\right)=\sum_{k=1}^{K}\omega_{k}g\left(x;m_{k},\sigma_{k}\right),$$
where $\omega_{i},m_{i},$ and $\sigma_{i}$ are shown in Table 1. To estimate the hyperparameters $\omega_{i},m_{i},$ and $\sigma_{i}$, the EM algorithm was used as described in Section 4. Table 1 shows the mean, standard deviation, and weight corresponding to the components of the mixture model. Figure 2 depicts the difference between the probability densities, $\Phi_{n}\left(x\right)$, and $\Xi_{EM}\left(\Phi_{n}\left(x\right)|g,\omega ,\mu ,\sigma ,K\right)$, where *K* = 1, 3, and 7. When comparing component *K*, raw data were similarly fitted to histograms when K = 3 and K = 7, and normal Gaussian modeling showed insufficient fitting for ESR, COX2, and CTSD (Figure 2). Commonly, the means and modes of the representative distributions existed near 0.5, and every distribution was skewed to the right.

**Gaussian distributions for queries:** To quantitatively compare the representative distributions corresponding to ESR, VDR, COX2, and CTSD with the query distributions, Kullback–Leibler divergence was introduced and calculated by building each distribution for each query.

For this purpose, $T_{M}\left[C_{l}\left(m\right),C_{l}\left(n\right)\right]$ of Equation (9) was used in a similar way to the described method for the representative distributions of the target classes. When a query was the *l*th ligand of $C_{l}\left(n\right)$, the *l*th column’s elements in the above matrix were used for the *l*th column vector, $\tau_{m}\left(m,n,l\right)$, as in
(13)$$\tau_{m}\left(m,n,l\right):=T_{M}\left[C_{l}\left(m\right),C_{l}\left(n\right)\right]E_{l}$$
where the values of $E_{l}$ for *j* = 1, 2, …, N were represented by the N × N matrices, for which the elements ${\left(E_{l}\right)}_{ij}$ satisfied
(14)$${\left(E_{l}\right)}_{ij}:=\left\{\begin{array}{l} 1,\;if\;i=j\\ 0,\;otherwise \end{array}\right.$$


Using the vector τmm,n,l from Equation (13), the following univariate probability densities, Φmnlxk, were defined as
(15)Φmnlxkδx:=ℙxk≤X=τmm,n,li≤xk+1
where the probability measure ℙ was derived from Equation (10).

Before obtaining the probability distribution, two assumptions were made. First, it was assumed that a distribution from one query was not a weighted sum of Gaussian distributions, but rather a simple Gaussian distribution. It was reasonable that a distribution from one query was simpler than the *Q*-distribution of a target class with 13,957 queries. Second, to estimate the parameters of the Gaussian distribution, ML estimation was chosen as a general method, in which
(16)$$\Xi_{ML}\left(\Phi_{mn}^{\left(l\right)}\left(x_{k}\right)|g,\omega ,\mu ,\sigma ,1\right)=g\left(x;\mu_{1},\sigma_{1}\right)$$
where *μ*_1_ and *σ*_1_ are hyperparameters and are maximized log likelihood functions for normal distribution, in other words,
(17)$$\left(\mu_{1},\sigma_{1}\right):=\arg\;\max_{\left(\mu ,\sigma \right)}\sum_{k=1}^{100}\frac{{\left(x_{k}-\mu \right)}^{2}}{\sigma^{2}}$$


Using definitions Equations (16) and (17), each query resulted in four distributions corresponding to the four classes (i.e., ESR, VDR, COX2, and CTSD). For example, when CHEMBL539392 was chosen as a query (*l*) among the ligands of ESR (Class 1), the distributions $\Phi_{11}^{\left(l\right)}\left(x_{k}\right),\Phi_{12}^{\left(l\right)}\left(x_{k}\right)$, $\Phi_{13}^{\left(l\right)}\left(x_{k}\right)$, and $\Phi_{14}^{\left(l\right)}\left(x_{k}\right)$ were obtained under the definitions of Equations (8) and (15). According to Equations (16) and (17), four representative Gaussian distributions of the query compound CHEMBL539392 were acquired from the column vector between CHEMBL539392 and 13,957 ligands of each class, which were
(18)$$\left\{\begin{array}{l} \Xi_{ML}\left(\Phi_{11}^{\left(l\right)}\left(x_{k}\right)|g,\omega ,\mu ,\sigma ,1\right)=g\left(x;0.24055,0.07472\right),\\ \Xi_{ML}\left(\Phi_{12}^{\left(l\right)}\left(x_{k}\right)|g,\omega ,\mu ,\sigma ,1\right)=g\left(x;0.21976,0.06466\right),\\ \Xi_{ML}\left(\Phi_{13}^{\left(l\right)}\left(x_{k}\right)|g,\omega ,\mu ,\sigma ,1\right)=g\left(x;0.24389,0.04857\right),\\ \Xi_{ML}\left(\Phi_{14}^{\left(l\right)}\left(x_{k}\right)|g,\omega ,\mu ,\sigma ,1\right)=g\left(x;0.21187,0.06631\right), \end{array}\right.\;\mathrm{for}\;k=0,1,\ldots ,99.$$


In the same way, univariate normal distributions were obtained of all of the query compounds in each class. Since the number of classes was four and there were 13,957 query compounds in each class, the Gaussian distributions $G\left(x;\mu_{1},\sigma_{1}\right)$, derived from $\Xi_{ML}\left(\Phi_{mn}^{\left(l\right)}\left(x_{k}\right)|g,\omega ,\mu ,\sigma ,1\right)$, presented the class number, either *m* or *n*, which was an integer between 1 and 4, and the query number, *l*, which was an integer from 1 to 13,957. As a result, the frequency distributions of the estimates, alongside the means ($\mu_{1}$) and standard deviations ($\sigma_{1}$), were described as shown in Figure 3 and Appendix A. ML estimation did not show any difference between self-query (m = n) and cross-query (m ≠ n) with regard to frequency. Even though cathepsin D (CTSD) showed a slightly lower mean than the other classes, self-comparison also showed a low mean, as shown in Figure 3. Regardless of whether a class or a query compound was used (self/cross), 3D similarity of ligand pairs within a class showed the mode near 0.6, thereby confirming the need for quantitative comparison between queries. Notably, the univariate probability distributions of 3D similarity did not discriminate between target class at all.

**Discrimination and K–L divergence**: In sequence, 3D similarity distributions of target classes and query compounds were quantitatively compared through K–L divergence calculations. First, the information describing specific Tanimoto–Jaccard coefficients, *x*, were defined as
(19)$$ln\left(\frac{\Xi_{ML}\left(\Phi_{mn}^{\left(l\right)}\left(x\right)|g,\omega ,\mu ,\sigma ,1\right)}{\Xi_{EM}\left(\Phi_{n}\left(x\right)|g,\omega ,\mu ,\sigma ,K\right)}\right)$$
from two probability density distributions, $\Xi_{ML}\left(\Phi_{mn}^{\left(l\right)}\left(x\right)|g,\omega ,\mu ,\sigma ,1\right)$ and $\Xi_{EM}\left(\Phi_{n}\left(x\right)|g,\omega ,\mu ,\sigma ,K\right)$, which were generated from a query compound and a class. Hence, following the expected value from the above information in Equation (19) with respect to one query compound, the K–L divergence,
(20)$$D\left(\Xi_{ML}\left(\phi_{mn}^{\left(l\right)}\left(x\right)|g,\omega ,\mu ,\sigma ,1\right)∥\Xi_{EM}\left(\phi_{n}\left(x\right)|g,\omega ,\mu ,\sigma ,K\right)\right)\;=\int \Xi_{ML}\left(\phi_{mn}^{\left(l\right)}\left(x\right)|g,\omega ,\mu ,\sigma ,1\right)ln\left(\frac{\Xi_{ML}\left(\phi_{mn}^{\left(l\right)}\left(x\right)|g,\omega ,\mu ,\sigma ,1\right)}{\Xi_{EM}\left(\phi_{n}\left(x\right)|g,\omega ,\mu ,\sigma ,K\right)}\right)dx$$
represented a measurement for the discrimination.

In a one-component GMM (*K* = 1), the K–L divergence between Gaussian distributions of every query and the *Q*-distributions (Table 1) are calculated; randomly chosen query compounds are described in Table 2. To show the calculation process in detail, CHEMBL539392 was chosen as an example. Using the above equation for Kullback–Leibler divergence between normal distributions,
(21)$$D\left(G\left(x;m_{i},\sigma_{i}\right)∥G\left(x;m_{j},\sigma_{j}\right)\right)=ln\left(\frac{\sigma_{j}}{\sigma_{i}}\right)+\frac{{\left(\sigma_{i}\right)}^{2}+{\left(m_{i}-m_{j}\right)}^{2}}{2{\left(\sigma_{j}\right)}^{2}}-\frac{1}{2}$$
where
(22)$$\left\{\begin{array}{l} G\left(x;m_{i},\sigma_{i}\right)=\Xi_{ML}(\phi_{1n}^{\left(1\right)}\left(x\right)|g,\omega ,\mu ,\sigma ,1)\\ G\left(x;m_{j},\sigma_{j}\right)=\Xi_{EM}(\phi_{n}\left(x\right)|g,\omega ,\mu ,\sigma ,1) \end{array}\right.$$


We obtained four K–L divergences corresponding to the queries of 2.1493, 4.6939, 2.0810, and 1.6354, respectively (see calculation procedure in the Appendix A. As shown in Table 2 and Appendix A, the K–L divergence of every query compound was not always the smallest value from their original targets, as annotated by ChEMBL DB. Even though a considerable number of query compounds showed that the K–L divergence resulting from an original target was smaller than values from other target classes, CHEMBL539392 of ESR, CHEMBL1163237 of COX2, and CHEMBL263810 of CTSD were considered to be less different than other targets, therefore giving a false prediction (Table 2). When we counted the query compounds that discriminated between the original targets and other targets from the 13,957 query compounds under the four classes via GMM (*K* = 1), the correct prediction numbers were 6300, 5200, 4100, and 6400 among each of the 13,957 queries from ESR, VDR, COX2, and CTSD, respectively. When applying GMM (*K* = 3) and (*K* = 7) for the *Q*-distributions, the true positive ratio decreased (ESR: 5100; VDR: 4500; COX2: 3200; CTSD: 4900 (*K* = 3); ESR: 4900; VDR: 4500; COX2: 3100; CTSD: 4800 (*K* = 7)).

In order to further evaluate the discriminative power of K–L divergence between target classes, an additional four classes as well as the four classes for BNDS-A were compared with the shared ligands in Table 3 and Appendix A. In Table 3, ritonavir (CHEMBL163) is a clinically approved drug on the HIV1 (human immunodeficiency virus type 1) protease as its primary target. Notably, ritonavir showed the distinct K–L divergence value to discriminate HIV1 with other targets. In addition, the result can rationalize why ritonavir cannot show a distinct difference between VDR and COX2. In contrast, myricetin (CHEMBL 164) showed very disappointing result with poor discrimination between K–L divergence values. However, when we checked every target of myrcetin, the natural compound did not show target specificity on any single protein to explain the result. The annotated activities were limited to the known targets (VDR: 31–40 μM, COX2: 100 μM, HSP90 13.5 μM in cell-based assay, TOP1: IC50 = 11.9 μg mL^−1^) in ChEMBL DB. Furthermore, despite the absent data on HIV1 of myrcetin, the flavonoid compound with multiple hydroxyl groups showed experimental activity on ubiquitin-specific protease having functional similarity (peptidase domain) with HIV1 to explain the K–L divergence value of 0.0393. In sequence, because reserpine (CHEMBL772), a clinically approved natural product, has target specificity on vesicular monoamine transporters with trivial activities on the annotated targets (VDR/COX2/TOP1), every target did not show a difference with untested targets (ESR/CPTD/HIV1). In addition, even though CHEMBL1813048 was the ligand of COX2 and TRPV4, K–L divergence could not support the finding. However, the result can be explained by the experimental data: (1) Ki against TRPV4 was more than 10 μM and (2) indirect regulation of COX2 was recorded through the Prostaglandin H2 receptor in ChEMBL DB. When compared with a 2D fingerprint based Top5 prediction of the additional target classes [36], our method can provide how much each query is quantitatively different with each target class from the raw data without any refinements such as assay, activity index, and duplicated ligands. This point is very important for investigating unprecedented drug scaffolds having weak activity out of the Top5 of a target class.

After the individual K–L divergence comparisons of each query, comparisons between the target classes were quantified. In sequence, the K–L divergence between the Gaussian distributions of 13,957 queries and the *Q*-distributions (K = 1, 3, and 7) for the four target classes were presented as a cumulative distribution, as seen in Figure 4, Figure 5, Figure 6 and Figure 7. To investigate the feasibility of the information, the following distribution was defined:
(23)ℙνlm=i for i=1,2,3,4,
where lm is the query number in class *m* and the random variable νlm represents a class number, so that
(24)νlm:=arg minn{D{ΞML(ϕmnlmx|g,ω,μ,σ,1)∥ΞEMϕnx|g,ω,μ,σ,1}|1≤n≤4,1≤lm≤13,957}


If the K–L divergence (Equation (20)) is an ideal measurement for discrimination between target classes, ($\nu \left(l_{m}\right)=i$) would satisfy the following conditions:
Necessary condition:
(25)ℙνlm=m≥maxi≠mℙ(νlm=i)
Sufficient condition: The feasibility index, Fm, is defined as
(26)Fm:=ℙνlm=m1−ℙνlm=m≥1



The above conditions implied a quantitative measurement for the discrimination. In particular, Fm in the sufficient condition represents the ratio between two probabilities (i.e., that a query compound belonged to a class of itself as well as belonging to other classes). A larger value of Fm indicated better feasibility or resolution of discrimination. Table 4 depicts the probability of the K–L divergence ℙνlm=i for 1≤i,m≤4, indicating that, except for example *m* = 3 where the class was COX2, the tested classes met the necessary conditions ℙνlm=m≥maxi≠mℙ(νlm=i) in Equation (25) with respect to the feasibility index in Equation (26), it was easier to distinguish a query compound in the CTSD class where *m* = 4 from every class except itself (Figure 8). When the feasibility index resulting from the GMM (*K* = 1) was compared with the index calculated from the GMM (*K* = 3) and (*K* = 7) for the *Q*-distributions, GMM (*K* = 1) showed superior feasibility for class discrimination using GMM (*K* = 3) or (*K* = 7), as shown in Table 4.

**Representative ligands for better discriminative predictions:** According to the results described in Figure 4, Figure 5, Figure 6, Figure 7 and Table 4, 3D similarity-based K–L divergence together with ℙνlm=m and F*_m_* showed discriminative power with regard to some query–class associations. The question therefore remains regarding the efficient use of the 3D-chemocentric approach under the current discriminative power, where it can be applied to investigate the novel pharmacology of an unprecedented compound. For this purpose, K–L divergence of an unprecedented compound should be calculated to compare known ligands and target classes. In detail, representative ligands within each target class were chosen for the comparison. For example, we selected four representative queries based on their Tanimoto–Jaccard coefficients (*x*), and K-L divergence value, namely, (1) *x* is the nearest to the mean of the *Q* distribution (GMM, *K* = 1), (2) *x* is the nearest to an outlier of the *Q* distribution (mean ± 2SD), (3) the range of K–L divergence between two target classes, and (4) the highest similarity with an unprecedented compound (Table 4). As an example, BNDS-A, a recently reported in-house compound [7], was used as the unprecedented compound due to the absence of ChEMBL DB. The first query compound close to the mean of the *Q* distribution showed a smaller K–L divergence than the other compounds (Table 5). The initial assumption and initial selection of the target class of BNDS-A (in other words, the selection of the *Q* distribution), resulted in a critical effect on the K–L divergence of BNDS-A as a query compound to predict the target class. When ESR was assumed as the initial target of BNDS-A, BNDS-A was more ESR ligand-like than CHEMBL558943 (at mean − 2SD for the ESR *Q* distribution) and CHEMBL604989 (which exhibited the biggest K–L divergence gap), and was less ESR-like than CHEMBL499809 (at the mean for the ESR *Q* distribution) and CHEMBL2 (at the mean + 2SD). Under the *Q* of ESR assumption, BNDS-A showed the lowest K–L divergence with the VDR ligands (0.0588 of VDR < 0.2116 of ESR), suggesting that BNDS-A was more VDR ligand-like than ESR ligand-like. When the initial target was transferred to VDR or COX2, BNDS-A showed the lowest K–L divergence required to satisfy the assumption (chosen *Q*). In all BNDS-A rows of Table 4, while the order of K–L divergence of BNDS-A (VDR < ESR < CTSD) was retained under the assumed every target class of BNDS-A, COX2 showed the lowest K–L divergence under only COX2 *Q* distribution and did not show consistent prediction. Therefore, BNDS-A was more VDR ligand-like than COX2 ligand-like. Experimentally, BNDS-A regulated the expression level of targets in a concentration-dependent manner (VDR > CTSD >> ESR) [7]. Notably, K–L divergence of 3D similarity distributions can be an additional comparison method of known methods to predict the target of a novel compound such as (1) the rank of 3D similarity score [7,15,16] or (2) p-value of one 3D similarity distribution [20]. Whenever achieving the relevance between a novel query and a target class is the aim, K–L divergence can be used for 3D-chemocentric informatics, as seen in the example of BNDS-A.

## 4. Materials and Methods

**Data collection:** All data, except for the in-house compound (BNDS-A), were extracted from the ChEMBL database (1. ESR, VDR, COX2, and CTSD: version 23 through KNIME community node, 2. HIV1, HSP90, TRPV4, and TOP1: version 25 through MySQL) [37]. Version 23 was available in both the ChEMBL community node of KNIME and in-house MySQL built from the dump file from ChEMBL ftp (ftp://ftp.ebi.ac.uk/pub/databases/chembl/ChEMBLdb/releases/). HIV1 protease, HSP90, TRPV4, and TOP1 data were chosen based on the literature [36] and downloaded from the ChEMBL 25 version.

**Conformational sampling:** Extracted compounds were converted from 2D structures into 3D conformation using Omega of the Openeye software [38] under the following conditions: (1) the MMFF94 force field excluding Coulomb interactions and the attractive part of Van der Waals interactions (option: *mmff94s_Trunc*) to retain the forces: bonding stretching, angle bending, stretch-bend interaction, out-of-plane bending at tricooridnate centers, torsion interaction, and the repulsive part of Van der Waals interactions; (2) 15 kcal/mol as the energy window; (3) hydrogen deletion from the input file fragment prior to the substructure search (option: *deleteFixHydrogens*); (4) permission to generate stereoisomers; and (5) maximum acceptable number of rotatable bonds of 25 [39]. Due to computational burden and space limitation to write similarity into a matrix during calculation at posterior work, 3D structures of every compound were merged into the structure files (file extension: sdf) according to target class, and 13,957 3D structures (with duplication due to different conformation) from the files were chosen via stratified sampling in KNIME to produce the dataset for similarity matrices as shown in Appendix A.

**Alignment method:** In order to align the 3D-structures of compound pairs, center of the mass was used [40]. In detail, it is reported that SIMPLEX algorithm for the alignment is already implemented in ROCS [15]. Shape Toolkit in the Openeye software [40,41] provides ‘*OEBOOrientation*’ used in *OEBestOverlay*. To optimize the alignment of each paired 3D structures, the starting point should be chosen before finding centers-of-mass of two conformers and *OEBestOverlay* uses an inertial frame alignment method to decide on starting positions by default. Under the default condition (‘*OEBOOrientation_Inertial*’), the first 3D structure (refmol in the python code in the Appendix A) was aligned by its principal moments of inertia, then the second structure (fitmol in the python code in the Appendix A) object was aligned in four positions with the primary and secondary moments of inertia in both possible directions. Therefore, the alignment of a compound pair (A, B) is approximately the same and absolutely not identical with the alignment (B, A).

**3D Descriptors:** In order to describe a molecular shape, atom-centered Gaussian sphere model was implemented in OE-MPI/ROCS and the Shape Toolkit [40,41]. OE-MPI, a kind of MPI (message passing interface), was also provided by Openeye for thread parallel calculation with a high number of CPUs. The Gaussian sphere model describing the 3D shape of compounds used the sum of Gaussian functions of individual heavy atoms except for hydrogen. *f* and *g* are characteristic functions to present the 3D atomic structure of each compound, *I*: self-volume overlaps of each entity, independent; *O*: the overlap between the two functions, dependent on orientation of two molecules.
(27)$$\mathrm{Shape}\left(f,g\right)=\sqrt{\int {\left[f\left(x,y,z\right)-g\left(x,y,z\right)\right]}^{2}dV}$$
$$\mathrm{Shape}{\left(f,g\right)}^{2}=\int {\left[f\left(x,y,z\right)\right]}^{2}dV+\int {\left[g\left(x,y,z\right)\right]}^{2}dV-2\int f\left(x,y,z\right)g\left(x,y,z\right)dV$$
$$\mathrm{Shape}\left(f,g\right)=I_{f}+I_{g}-2O_{f,g}$$
$$\text{Jaccard–Tanimoto coefficient of Shape}\left(f,g\right)=\frac{O_{f,g}}{I_{f}+I_{g}-O_{f,g}}$$


Color features of every query were generated under the default algorithm of the Shape Toolkit. Color features were defined by pharmacophore types (H-bond donor, H-bond acceptor, negative charge, positive charge, hydrophobic, and ring) in a color force field (*Implicit Mills Dean*) and color atoms were described by Gaussian functions as being relatively hard with a steep gradient.

**3D Similarity matrix:** The Jaccard–Tanimoto coefficient of two features, shape and color were calculated, combined, and written into 3D similarity matrices using the functions in the supplementary python script [42].
-OEOverlay(): optimization of the alignment(overlap) between query and database-OEBestOverlayScoreIter(): sorting all scores to highest Tanimoto coefficient before writing similarity score into an empty matrix.


In this study, while the dimension of 3D similarity matrices for *Q* distributions (GMM) was 13,957 by 13,957, the dimension of 3D similarity matrices for query distributions (ML estimation) was 1 by 13,957. Every sampled compound of four target classes (13,957 conformers x four target classes) was used as the query to show the performance of K–L divergence. The BNDS-A compound is only one query not existing in any target class.

**Script for K–L divergence.** In order to realize (1) the GMM model, (2) the ML estimation, and (3) K–L divergence, python scripts were written using python libraries such as pandas [43], numpy [44], and scipy [45] under anaconda installation [46], so that every code was uploaded to GitHub [47].

## 5. Conclusions

We developed a quantitative method comparing query compounds to target classes. The discriminative comparison was achieved by K–L divergence of 3D similarity distributions. The distributions were generated from 3D structures (sampled multi-conformers) with target annotation and optimized with parameters to best fit to frequent histograms. The feasibility index, F_m_, and the probability, P(ν(l_m_) = i), derived from the K–L divergence demonstrates the discrimination of queries against target classes. The feasibility index resulting from the GMM (*K* = 1) showed better feasibility for class discrimination than the GMM (*K* = 3) and (*K* = 7). Among the target classes, CTSD showed the most desirable feasibility and COX2 was the least desirable target for chemocentric informatics. K–L divergence comparison of an unprecedented compound, BNDS-A showed the consistent order of K–L divergence of BNDS-A (VDR < ESR < CTSD) under different target assumptions of BNDS-A so that our method is applicable for discriminative predictions of unknown query compounds in chemocentric informatics. This study will contribute to 3D chemocentric target deconvolution for unprecedented drug scaffolds. In the recent future, this quantitative method should be further studied with regard to the field of chemical optimization between the chemical space and pharmacological space.

## Figures and Tables

**Figure 1 ijms-21-04208-f001:**
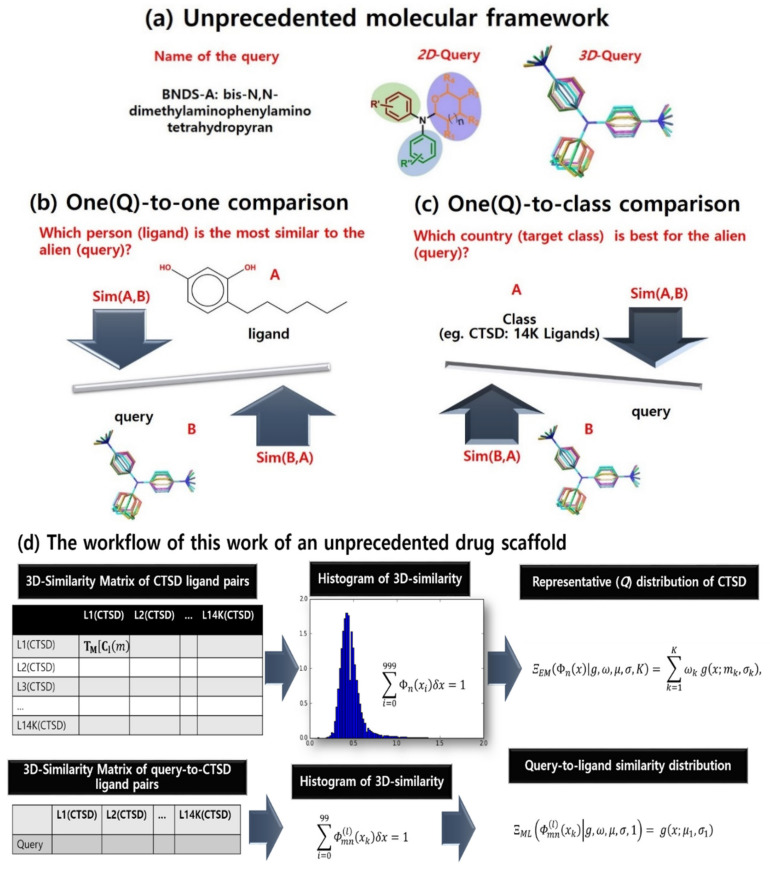
The problem definition of 3D chemo-centric screening. (**a**) BNDS-A as a new molecular framework. (**b**) The role of chemical similarity in virtual screening. (**c**) The role of chemical similarity in chemo-centric retro-virtual screening. (**d**) The workflow of this work of an unprecedented drug scaffold.

**Figure 2 ijms-21-04208-f002:**
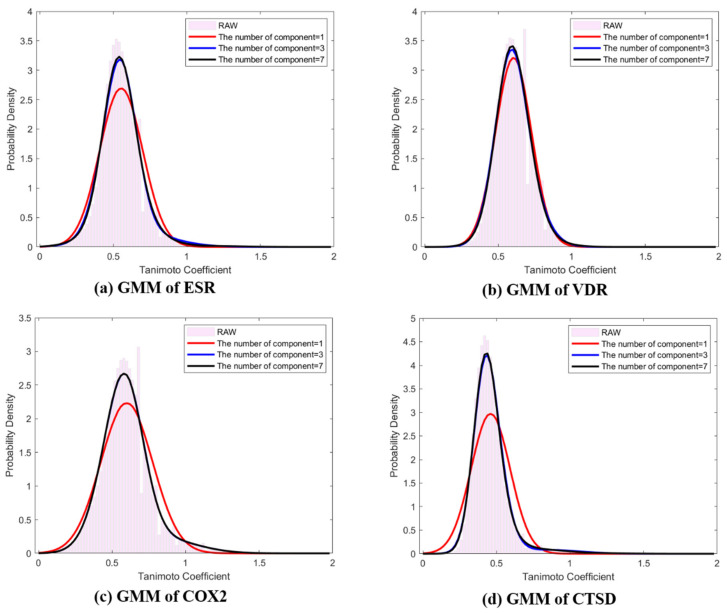
Representative distributions (*Q*-distributions) of target classes using EM based Gaussian mixture model ($\Xi_{EM}\left(\Phi_{n}\left(x\right)|g,\omega ,\mu ,\sigma ,K\right)$ of ligand pair similarity. (**a**) *Q*-distribution of ESR; (**b**) *Q*-distribution of VDR; (**c**) *Q*-distribution of COX2; (**d**) *Q*-distribution of CTSD. The red line: GMM *K* = 1, blue line: GMM *K* = 3, black line: GMM *K* = 7, pink bar: histogram of raw data.

**Figure 3 ijms-21-04208-f003:**
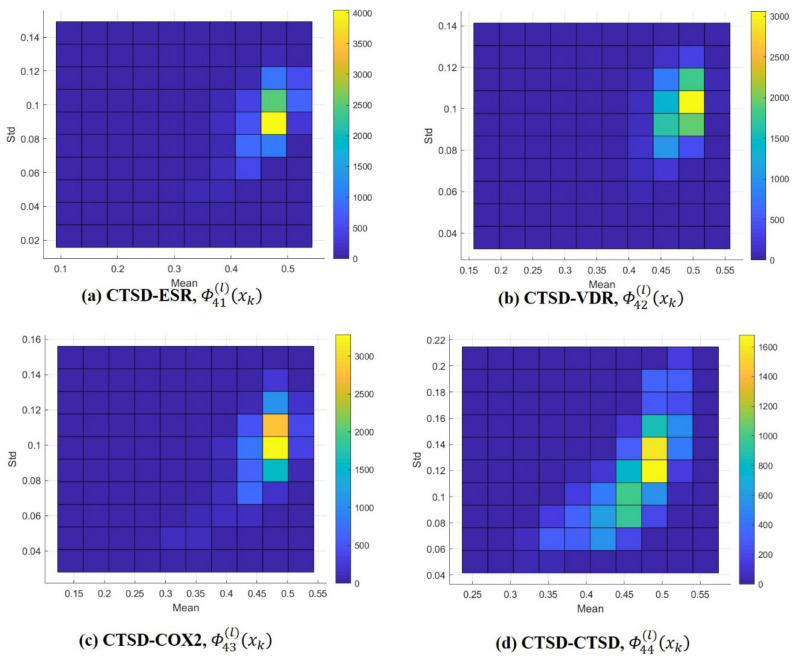
Frequency distributions of $\Xi_{ML}\left(\Phi_{4n}^{\left(l\right)}\left(x_{k}\right)|g,\omega ,\mu ,\sigma ,1\right)$ estimates ($\mu_{1}$ and $\sigma_{1}$). Query (*l*) ∈ CTSD (class = 4). (**a**) CTSD-ESR, (**b**) CTSD-VDR, (**c**) CTSD-COX2, and (**d**) CTSD-CTSD. * The color bars (right side of the distribution) indicate frequency (e.g., yellow in 3(a) represents 3500 to 4000 queries, the mean of the ML estimates varied from 0.45 to 0.5 and their standard deviation varied from 0.08 to 0.1 in the standard).

**Figure 4 ijms-21-04208-f004:**
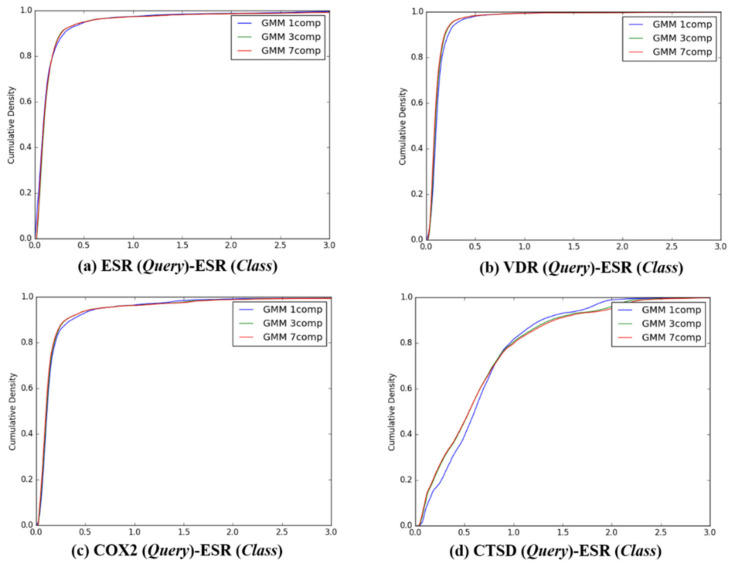
The cumulative densities of K–L distance between *Q*-distribution (Target class: ESR) and queries. *X*-axis: K–L divergence, *Y*-axis: cumulative density; *Q*-distribution of ESR through GMM and the distribution of queries were calculated. (**a**) ESR(Query)-ESR(Class), (**b**) VDR(Query)-ESR(Class), (**c**) COX2(Query)-ESR(Class), and (**d**) ESR(Query)-ESR(Class).

**Figure 5 ijms-21-04208-f005:**
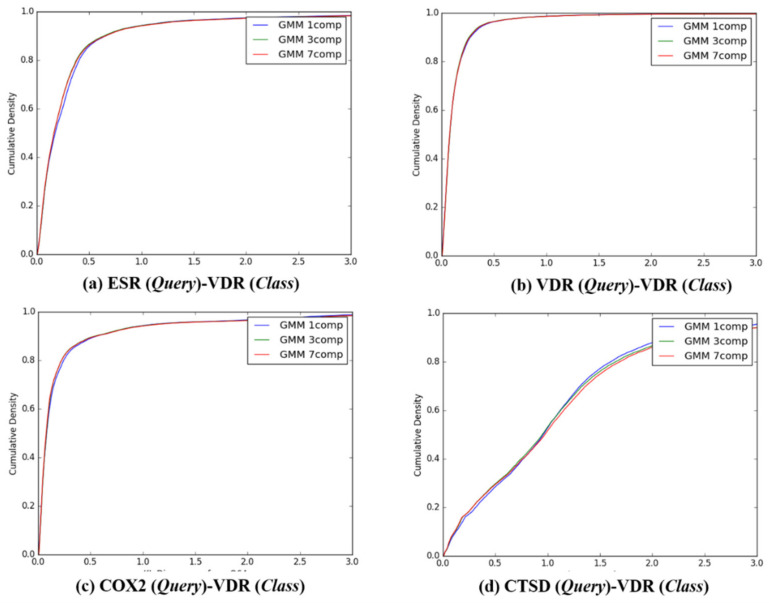
The cumulative densities of K–L distance between *Q*-distribution (Target class: VDR) and queries. *X*-axis: K–L divergence, *Y*-axis: cumulative density; *Q*-distribution of VDR through GMM and the distribution of queries were calculated. (**a**) ESR(Query)-VDR(Class), (**b**) VDR(Query)-VDR(Class), (**c**) COX2(Query)-VDR(Class), and (**d**) ESR(Query)-VDR(Class).

**Figure 6 ijms-21-04208-f006:**
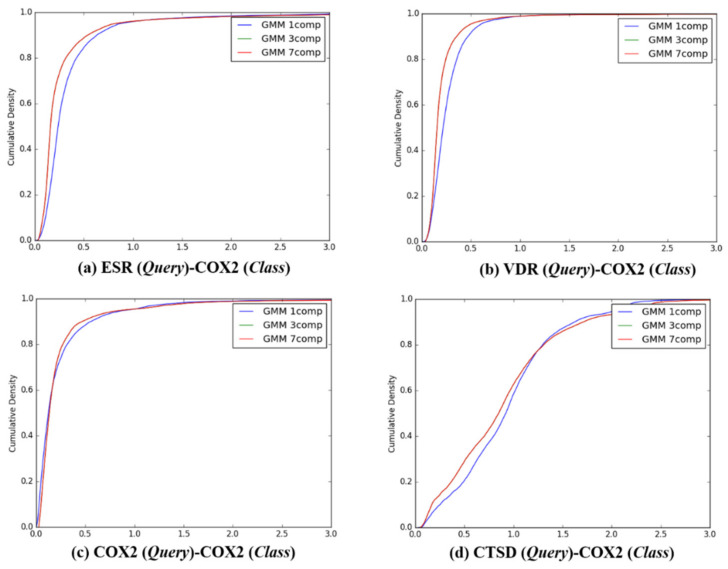
The cumulative densities of K–L distance between *Q*-distribution (Target class: COX2) and queries. *X*-axis: K–L divergence, *Y*-axis: cumulative density; *Q*-distribution of COX2 through GMM and the distribution of queries were calculated. (**a**) ESR(Query)-COX2(Class), (**b**) VDR(Query)-COX2(Class), (**c**) COX2(Query)-COX2(Class), and (**d**) ESR(Query)-COX2(Class).

**Figure 7 ijms-21-04208-f007:**
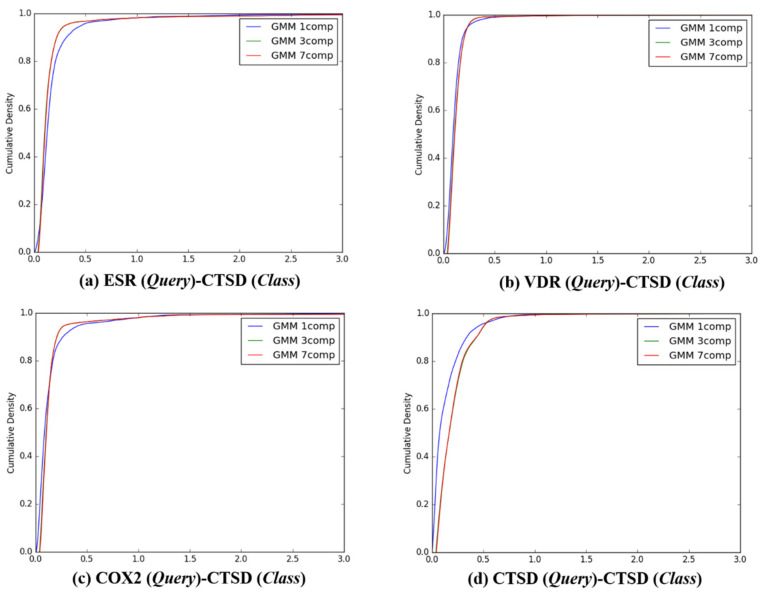
The cumulative densities of K–L distance between *Q*-distribution (Target class: CTSD) and queries. *X*-axis: K–L divergence, *Y*-axis: cumulative density; *Q*-distribution of CTSD through GMM and the distribution of queries were calculated. (**a**) ESR(Query)-CTSD(Class), (**b**) VDR(Query)-CTSD(Class), (**c**) COX2(Query)-CTSD(Class), and (**d**) ESR(Query)-CTSD(Class).

**Figure 8 ijms-21-04208-f008:**
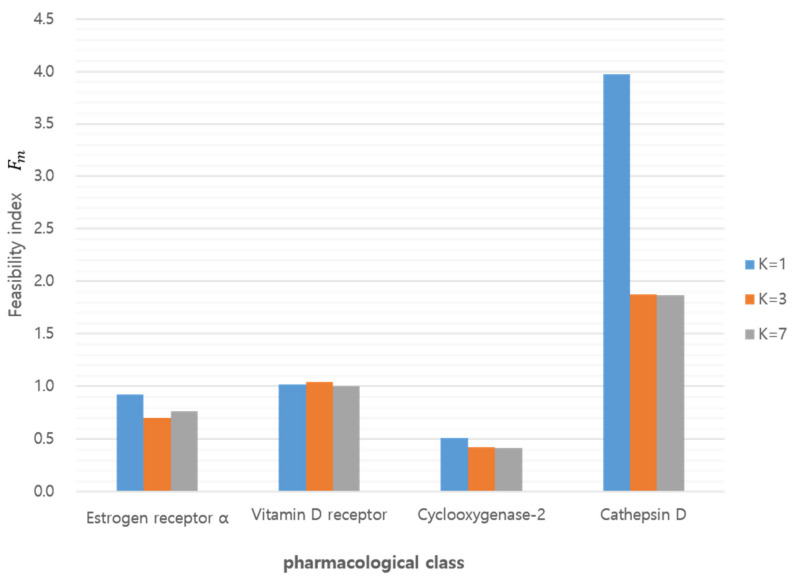
Feasibility index according to target class and GMM component (*K*).

**Table 1 ijms-21-04208-t001:** Hyperparameters of *Q* distributions for target classes.

**GMM**	**ESR**		**VDR**		**COX2**		**CTSD**	
No(i)	$m_{i}$	$\sigma_{i}$	$m_{i}$	$\sigma_{i}$	$m_{i}$	$\sigma_{i}$	$m_{i}$	$\sigma_{i}$
1	0.5483	0.1458	0.5981	0.1224	0.5941	0.1758	0.4560	0.1320
**GMM**	**HIV1**		**HSP90**		**TRPV1**		**TOP1**	
No(i)	$m_{i}$	$\sigma_{i}$	$m_{i}$	$\sigma_{i}$	$m_{i}$	$\sigma_{i}$	$m_{i}$	$\sigma_{i}$
1	0.419	0.123	0.614	0.206	0.667	0.176	0.510	0.222

**Table 2 ijms-21-04208-t002:** K–L divergence of randomly chosen queries between *Q* distributions and the distributions of queries.

Class	Query	K–L Divergence			
		ESR	VDR	COX2	CTSD
ESR	CHEMBL	2.6310	5.2420	2.9952	1.9426
	539392				
	CHEMBL	0.0223	0.1144	0.0685	0.0363
	193280				
	CHEMBL	0.0564	0.1847	0.1638	0.2186
	443605				
VDR	CHEMBL	0.0658	0.0107	0.0795	0.0637
	7162				
	CHEMBL	0.0488	0.0420	0.2391	0.0682
	1322390				
	CHEMBL	0.0983	0.0849	0.3748	0.1003
	1452735				
COX2	CHEMBL	0.4773	0.7264	0.4693	0.2694
	1163237				
	CHEMBL	0.0811	0.0436	0.0326	0.0490
	127560				
	CHEMBL	0.0704	0.0417	0.0684	0.0724
	271614				
CTSD	CHEMBL	0.0889	0.0146	0.2667	0.1014
	263810				
	CHEMBL	0.6800	1.0065	0.9193	0.1174
	252655				
	CHEMBL	0.5331	0.8771	0.8109	0.0766
	436438				

**Table 3 ijms-21-04208-t003:** K–L divergence of ligands shared with eight target classes *.

Query	Targets	ESR	VDR	COX2	CTSD	HIV1	HSP90	TRPV4	TOP1
CHEMBL	VDR/COX2/HIV1	1.2649	2.2088	1.6702	0.6982	**0.3587**	1.6040	1.9256	1.2754
163									
(RITONAVIR)									
CHEMBL	VDR/COX2/HSP90/TOP1	0.0718	**0.0526**	0.1148	0.0475	0.0393	0.1655	0.5684	0.0915
164									
(MYRICETIN)									
CHEMBL	ESR/VDR/COX2/TOP1	**0.3075**	0.4963	0.6972	0.2792	0.1685	0.8460	0.7630	0.5009
772									
(RESERPINE)									
CHEMBL	COX2/TPRV4	0.2385	0.3053	**0.4731**	0.2322	0.1704	0.6374	0.6669	0.5810
1813048									

* The smallest K–L divergence value among the experimentally tested targets of each query is presented in bold.

**Table 4 ijms-21-04208-t004:** The description on ℙνlm=i and Fm according to the number of components of Gaussian Mixture Model K, and the class νlm of queries lm
^a^.

**K = 1**	ℙνlm=i	Fm ^b^
Class of representative distributions i
ESR	VDR	COX2	CTSD
Class $\nu \left(l_{m}\right)$ of queries $l_{m}$	ESR	**0.4623**	0.2172	0.0082	0.3123	0.9272
VDR	0.1116	**0.5101**	0.0054	0.3729	1.0205
COX2	0.0882	0.3216	0.2046	0.3856	0.5071
CTSD	0.0051	0.0489	0.0057	0.9404	3.9718
**K = 3**	ℙνlm=i	Fm ^b^
Class of representative distributions i
ESR	VDR	COX2	CTSD
Class $\nu \left(l_{m}\right)$ of queries $l_{m}$	ESR	0.3289	0.2616	0.0725	0.3370	0.7001
VDR	0.1653	0.5199	0.0517	0.2631	1.0406
COX2	0.1024	0.4922	0.1534	0.2520	0.4257
CTSD	0.1348	0.0741	0.0128	0.7783	1.8738
**K = 7**	ℙνlm=i	Fm ^b^
Class of representative distributions i
ESR	VDR	COX2	CTSD
Class $\nu \left(l_{m}\right)$ of queries $l_{m}$	ESR	0.3669	0.2553	0.0713	0.3065	0.7613
VDR	0.2164	0.5005	0.0476	0.2356	1.0009
COX2	0.1387	0.4891	0.1477	0.2245	0.4164
CTSD	0.1437	0.0705	0.0084	0.7775	1.8691

^a^ This table represents the feasibility of discrimination depending on the number of components in GMM, K, and the class ν(l_m_) of queries l_m_. ^b^ The larger $F_{m}$, the better performance of discrimination between one class and others. Estrogen receptor alpha = ESR, Vitamin D receptor = VDR, Cyclooxygenase-2 = COX2, Cathepsin D = CTSD.

**Table 5 ijms-21-04208-t005:** The comparison between representative queries and unprecedented drug BNDS-A as a query.

Class	Query	Selection Type	Max. of K–L Divergence			
			ESR	VDR	COX2	CTSD
**ESR**	CHEMBL 499809	Mean of *Q*	0.0363	0.1991	0.1611	0.2772
	CHEMBL 2	(Mean + 2SD) of *Q*	0.1180	0.1001	0.1547	0.0883
	CHEMBL 558943	(Mean − 2SD) of *Q*	2.7919	5.2859	2.9632	2.0501
	CHEMBL 604989	Biggest gap of K–L divergence	6.2458	10.9899	6.1578	5.4983
	CHEMBL 292033	Highest Similarity with BNDS-A	0.0298	0.2570	0.2096	0.1082
	BNDS-A	Unknown	0.2116	**0.0588**	0.1139	0.9704
**VDR**	CHEMBL 7463	Mean of *Q*	0.0237	0.0442	0.1446	0.1262
	CHEMBL 603	(Mean + 2SD) of *Q*	0.0999	0.2738	0.1257	0.0655
	CHEMBL 1116	(Mean − 2SD) of *Q*	1.2883	2.1898	1.6169	0.4702
	CHEMBL 486541	Biggest gap of K–L divergence	4.2675	7.2936	3.9890	3.3430
	CHEMBL 62136	Highest Similarity with BNDS-A	0.2090	0.1854	0.4785	0.1086
	BNDS-A	Unknown	0.2859	**0.0864**	0.1888	1.0807
**COX2**	CHEMBL 1201356	Mean of *Q*	0.0963	0.1054	0.2187	0.0948
	CHEMBL 16516	(Mean + 2SD) of *Q*	0.1445	0.1172	0.0385	0.1205
	CHEMBL 1171450	(Mean − 2SD) of *Q*	3.2143	5.5460	3.1399	2.4262
	CHEMBL 1171454	Biggest gap of K–L divergence	4.4382	7.8994	4.1848	4.1940
	CHEMBL 942	Highest Similarity with BNDS-A	0.1285	0.0546	0.09018	0.06225
	BNDS-A	Unknown	0.6987	0.65378	**0.2273**	2.0276
**CTSD**	CHEMBL 263810	Mean of *Q*	0.0850	0.0113	0.2512	0.1038
	CHEMBL 504438	(Mean + 2SD) of *Q*	0.6941	1.1751	1.1002	0.3305
	CHEMBL 567893	(Mean − 2SD) of *Q*	3.5366	6.1606	3.5399	2.0713
	CHEMBL 567893	Biggest gap of K–L divergence	3.5684	6.1606	3.5399	2.0713
	CHEMBL 387576	Highest Similarity with BNDS-A	0.0835	0.1467	0.0952	0.0129
	BNDS-A	Unknown	**0.0556**	0.26421	0.2092	0.087

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
