# Peer review of "Comparing a Query Compound with Drug Target Classes Using 3D-Chemical Similarity"

_ijms, 2020, doi:10.3390/ijms21124208_

Round 1

Reviewer 1 Report

I appreciate the efforts of authors in improving the manuscript. The work and the manuscript have clearly improved from version 1. The only recommendation I have is to reduce the total number of figures in the main text. If possible, please move some of them to the supplementary material.

Author Response

I, along with my co-authors, revised our manuscript based on the review comments. Now I respond to the comments point by point. Kindly ask you to read the response. For facile reading, we showed marked track-changed (blue color) in manuscript and line number of a revised part in this letter.

Peer-reviewer 1

[Main concern] I appreciate the efforts of authors in improving the manuscript. The work and the manuscript have clearly improved from version 1. The only recommendation I have is to reduce the total number of figures in the main text. If possible, please move some of them to the supplementary material.
Ans. I transferred 3 figures to show Frequency distributions of MLL estimates into Supplementary Information and remained one example in the main text.

Reviewer 2 Report

IJMS_822442

This manuscript presents an in-depth description of a computational method aimed at comparing query compounds to target classes using Kullback-Leibler divergence on 3D chemical similarity. I am not an expert in the field, but I am aware of the need to improve virtual screening accuracy of chemical fragments in drug discovery, and this work aids in fulfilling this need.

The paper uses a set of 4 test compounds and reports their results, which are documented in 11 Figures and 5 Tables displayed in the main text, plus additional items in the Supplementary Material. The text is exhaustive in describing the methods used and the theory behind the calculations. Python codes are also included, and have been deposited in GitHub. Thus, the authors have extensively presented their approach.

My main concern is whether a reader will be sufficiently motivated to view all the displayed items, or whether a selection of salient data shown in 3-5 Figures at most would not suffice. In general, figures appear more like a data dump than a mean to convey key information.

My other concern is about the English style. There are innumerable mistakes, many of which should have been avoided in the first place. This conveys a negative impression on the reviewers and if not corrected, will turn off a reader. I have listed several of them but there may be others.

Specific comments

Line 11 – frameworks that are

Line 28 - additional method for: (1) the …

Line 37 - in virtual screening (VS;  Figure 1B) [1-2].

Line 38 - Figure 1A should be cited before Figure 1B.

Line 59 – Figure 1 does not have a panel D.

Line 77 - To our knowledge, such measurements have NOT BEEN reported in cheminformatics. Is this what the authors want to say?

Line 80 - However, even though the number of studies using 3D similarity IS enormous [15, 16], 3D similarity …

Line 82 - in THE literature

Line 83 - algorithm-based classifiers

Line 84 - 18]. Almost ALL (is this what the authors mean?) classifiers do no use only chemical similarity but also other descriptors [18].

Line 85 – Although several studies treated 3D similarity distribution [5,19,20], the distribution COMPRISED every compound instead of groups according to target [5, 19].

Line 90 – optimized, in spite of the small …

Line 94 - of target classes (Figure 1(d)). Also, a, b, c, d should be the same size.

Lines 96-97 - row of Figure 1(d)).

Line 118 – The eq numbers should all be either italic or not italic, not both

Line 120 – “or” is already on the previous line.

Line 135 – no comma at the end.

Line 139 – what is the first symbol?

Line 140 – I did not see eq. 2.9.

Line 147 – should p? be in eq. 2.7?

Line 169 - GMM allowed to obtain a representative

Line 183 - proved by a stepwise approach

Figure 2 – I would suggest making this figure a 2x2 panels; I do not see the abbreviations and a, b, c and d.

Line 249 - assumptions were made.

Line 326 - Supplementary Table 3

Line 331 - prediction (Table 2).

Line 334 – respectively.

Line 339 - in Table 3 and Supplementary Table 2. In Table 3, ritonavir

Line 340 - approved drug on HIV1(Human Immunodeficiency Virus type 1) protease as…

Line 343 – myrcetin -> do you mean myricetin?

Line 346 - annotated activities WERE limited – please use the Geek letter mu for micromolar

Line 354 - could not support the finding?

Line 356 - was recorded through Prostaglandin H2

Line 357 – fingerprint

Line 358 - how much each query IS quantitatively

Line 359 – Thins -> This

Line 360 - very far from Top5. What do you mean by very far.

Line 386 - of discrimination.

Line 420 - and Table 4

Line 426 - four representative

Line 443 - rows of Table 4,

Line 458 – The 23 version

Line 460 – I did not understand the meaning: do you mean “To get additional datasets with different four-target data, additional four-target data were collected from the literature [36].

Line 474 - Supplementary Table 1.

Line 510 - as query to show

Line 514 - to GitHub

Supplementary Information

Page 1 - Summary of datasets

  1. Figures 1 to 4

Eq. S1.4 has an undefined symbol

Eq. S1.7 – please check that the integral is not missing any symbol

Please check that Eq. 3.15 is not missing any symbol.

  1. Table 1. Summary of datasets.

Heat shock protein 90 (HSP90)

Abbreviation and terminology list

#1 - have similar properties so …

#2 - However, ‘novel’ cannot mean ‘first existing’

#4 - is a method that analyzes compositional

#6 – commonimity. Is this an English word? Do you mean commonality?

#8 - ChOS consists of four

Author Response

Response letter

To editorial board of International Journal of Molecular Sciences and peer-reviewers,

I, along with my co-authors, revised our manuscript based on the review comments. Now I respond to the comments point by point. Kindly ask you to read the response. For facile reading, we showed marked track-changed (blue color) in manuscript and line number of a revised part in this letter.

[Main concern] is whether a reader will be sufficiently motivated to view all the displayed items, or whether a selection of salient data shown in 3-5 Figures at most would not suffice. In general, figures appear more like a data dump than a mean to convey key information.

Ans. I transferred 3 figures to show Frequency distributions of MLL estimates into Supplementary Information and remain one example in the main text. I think that now essential Figures exist in the main text. Figure 1 introduces the whole study. Figure 2 justify the GMM model. Figure 3 described calculated MLL estimate. Figure 4-7 should be presented to show K-L divergence pattern. Figure 8 visualized Fm value.

[Q1-English revision]

The manuscript received English editing service to remove the issue. English editing was conducted in MDPI English service (with the attached certificate). Every mentioned misspelling or error was revised and marked track-changed (blue color) in manuscript.

Q1-1. Line 360 - very far from Top5. What do you mean by very far.

Ans1-1. out of Top5 of a target class.

Q1-2. Line 460 – I did not understand the meaning: do you mean “To get additional datasets with different four-target data, additional four-target data were collected from the literature [36].

Ans1-2. HIV1 protease, HSP90, TRPV4, and TOP1 data were chosen based on the literature [36] and downloaded from ChEMBL 25 version.

[Q2-Figures]

Figure 1 and Figure 2 were revised according to the points: “figure 2 a 2x2 panels”, “Figure 1 a, b, c, d should be the same size:, “Figure 1A should be cited before Figure 1B”)

[Q3-Mathmatic Equation]

Q3-1. Line 139 – what is the first symbol? Eq. S1.4 has an undefined symbol.

Ans3-1.   is the operator for expected value having probability density g(x;m_i, σ_i ), (and so it implies that Kullback-Leibler divergence is expected value of difference between two probabilities. ) 

Q3-2. Line 147 –should p? be in eq. 2.7? 

Ans3-2. Instead of p?, we used

Q3-3. Eq. S1.7 – please check that the integral is not missing any symbol

Ans3-3. The range of integral was expressed. (S1-7)

Q3-4. Please check that Eq. 3.15 is not missing any symbol.

Ans3-1.  are functional operators for EM-optimized GMM and MLE, respectively. They already were defined in Theoretical background of our main text.

[Q4-Supplementary/ Abbreviation and terminology list]

Every mentioned misspelling or error was revised and marked track-changed (blue color) in the files.

This manuscript is a resubmission of an earlier submission. The following is a list of the peer review reports and author responses from that submission.

Round 1

Reviewer 1 Report

The authors present a method based on 3D chemical similarity that is useful to compare a
ligand with a set of ligands belonging to a target class. Analysis was performed using ligand
sets belonging to four pharmacological targets. Quantitative metrics that estimate the
relationship between a ligand and target sets were presented with a use-case. The authors
also make the code used for analysis publicly available on GitHub.
While the research presented is interesting, the manuscript suffers from poor language and
presentation of results. A significant portion of the manuscript is dominated by mathematical
notations which might make it hard to interpret the results. Also, the authors do not provide a
comparison with previous 3D or 2D similarity-based methods that try to address the
same/similar question. I believe that the current work could be further improved by not only
simplifying the presentation of methods and results but also validating the method by
comparing it to previously proposed method(s). Please find below my recommendations for
improving different sections of the manuscript.
Abstract :
Line 13: it should be ‘ the alignment method’
Line 14: what do the authors mean by ‘metric to show limited discriminative power’?
Line 14: it should be ‘In addition to these / the four factors’
Line 18: authors should check with the journal guidelines if acronyms are allowed in the
abstract (there are several other acronyms used in the abstract)
Line 25: it is better to write it as ‘ 14000 ligands’
Line 25: it would be good to mention the source(s) of these ligands (e.g. A total of 14000
ligands were sampled in a stratified fashion from XYZ database/resource… )
Overall, the abstract misses out on clearly stating the outcomes of the study (e.g. ‘ Feasibility
index and the probability from K-L divergence could be successfully used to assess the
chemocentric relationship between the target classes’ ). Furthermore, a short statement on
whether this approach was compared to previous methods (2D/3D) from the literature would
add more value.
Introduction :
Line 35: considering that this is its first appearance in the manuscript (excluding the
abstract), 3D should be written as ‘ three-dimensional (3D) ’ and same should be the case
with 2D
Line 42-46: please rephrase these sentences
Line 48-52: these are major claims, to support which the authors might need to cite
milestone studies from the literature that have led to these conclusions. alternatively, or in
addition, the authors could elaborate on the limitations of 2D similarity (e.g. the aspects that
it does not take into consideration)
Line 62-63: please rephrase this sentence
Line 67: ‘To achieve the goal, …’
Line 68: please rephrase the last sentence of the paragraph
Line 71: ‘two types of similarity
Line 73: ‘coefficient was calculated for (1) all query-to-ligand pairs and (2) all ligand
pairs within each target class ’
Line 74,75: ‘similarity was transformed…’
Line 77-78: this sentence could be better presented (e.g. ‘ estimation of K-L divergence of a
query-to-target refers to quantification of …. between a query ligand a target class. )
Line 79: please rephrase the complete sentence
Line 81: ‘Furthermore,’ could be used instead of ‘In sequence’
To summarize, the introduction section is poorly constructed. The authors are recommended
to address the above issue and make sure the language is clear and consistent.
Materials and methods :
While I like the idea of providing a detailed theoretical background for the techniques
employed, I felt that this part is disconnected to the study methodology. I encourage the
authors to combine this with the materials and methods section and consider limiting the
number of mathematical expressions used to make it simple for the readers.
Version 26 of ChEMBL has been available since a few weeks and version 25 for a while
now. Is there a reason why version 23 was used?
The readers would also benefit from a table that summarizes the datasets used in the study
(e.g. the number of compounds from each dataset or target). Was there any overlap in terms
of ligands between these four datasets? If yes, how were they treated?
Also, it is a well-known fact that the data from ChEMBL has the drawback of experimental
variability since they extract data from different sources. Considering this, have the authors
used any sort of preprocessing in order to clean the data for each target? How did they deal
with duplicate structures and measurements? Did they standardize the chemical structures?
What do the authors mean by ‘combined 3D structures? (Line 422)
Results and discussion :
The labels a, b, c and d in Figure 6 should be CTSD-ESR, CTSD-VDR, CTSD-COX-2 and
CTSD-CTSD, respectively.
The figures look fine, but the tables are poorly organized with few of the rows containing
large chemical structures. The authors could omit the structures from the tables and present
them elsewhere (e.g., supporting information?)
This section needs to be better presented in terms of language and organizing the figures
and tables. Also, what I currently do not see is the discussion around the results. Are there
similar efforts in the literature? If none, how does this new 3D method compare to existing
2D methods that try to establish relation between targets based on their ligand sets? Adding
this comparison would add more value to this work.
Overall, although I recognize the efforts and the value of the work, the manuscript in its
current form may not be suitable for publication at IJMS. I would like to see an improved
version from the authors before it could be accepted for publication.

Author Response

Response letter

To editorial board of International Journal of Molecular Sciences and peer-reviewers,

I, along with my co-authors, revised our manuscript based on the review comments. Now I respond to the comments point by point. Kindly ask you to read the response. For facile reading, we showed marked track-changed (red color) in manuscript and line number of a revised part in this letter.

Peer-reviewer 1

[Q1-Revision of English, Rephrase, Acronyms of Abstract]
According to the suggestion of reviewer 1, abstract was revised.

Abstract was re-written based on three peer-reviewers’ comments.

English editing was conducted in MDPI English service (with the attached certificate).

Mentioned parts with line number were revised with red-color in our revision.

List of acronyms was added in the revision.

[citation of milestone studies from the literature] Kindly ask you to see lines No. 87 to 102 with references [5-20]. In addition, you can see Ans3-4 with the references.

[Q2-Materials and methods]:

Q2-1. Version 26 of ChEMBL has been available since a few weeks and version 25 for a while
now. Is there a reason why version 23 was used?
Ans2-1. The version 23 was chosen due the availability in ChEMBL community node of KNIME and in-house MySQL built from the dump file from ChEMBL ftp. I added the description into material and methods section.

Q2-2. The readers would also benefit from a table that summarizes the datasets used in the study.

Ans2-2.The table that summarizes the datasets used in the study exist in supplementary Table 1.

Q2-3. Was there any overlap in terms of ligands between these four datasets? If yes, how were they treated?

Ans2-3. When the duplication of the ligands was checked using ‘chembl ID of compound’ in KNIME. With the comment, we mentioned overlapped dataset information in supplementary Table2. If we delete duplicated data, we can expect improved performance. However, because distribution follow parametric statistics, original data should be retained to prevent any distortion.

Q2-4. Also, it is a well-known fact that the data from ChEMBL has the drawback of experimental variability since they extract data from different sources. Considering this, have the authors used any sort of preprocessing in order to clean the data for each target? How did they deal with duplicate structures and measurements? Did they standardize the chemical structures? What do the authors mean by ‘combined 3D structures? (Line 422)

Ans2-4. In order to escape any duplication, ‘chembl ID of compound’ was used as a unique key and any assay ID, document ID, or activity ID was not used. Initial structure of every compound was ‘smiles’ from ChEMBL and 3D structure were generated in Omega of Openeye Software. ‘combined 3D structures? (Line 422) means the combination of sdf file in Knime.

[Q3- Results and discussion]: The labels a, b, c and d in Figure 6 should be CTSD-ESR, CTSD-VDR, CTSD-COX-2 and CTSD-CTSD, respectively. The figures look fine, but the tables are poorly organized with few of the rows containing large chemical structures. The authors could omit the structures from the tables and present them elsewhere (e.g., supporting information?) This section needs to be better presented in terms of language and organizing the figures and tables. Also, what I currently do not see is the discussion around the results.  Are there similar efforts in the literature? If none, how does this new 3D method compare to existing 2D methods that try to establish relation between targets based on their ligand sets? Adding this comparison would add more value to this work.

Ans3-1. The caption of Figure 3-6 was revised for better readability.

Ans3-2. Figures in the table 2 are moved to supplementary table 3.

Ans3-3. [the discussion around the results] Carefully I suggest the revised conclusion; Line No. 550-565 describe our results. You can also see line No. 385-390, 391-417, 436-444, and 447-482.

Ans3-4. [similar efforts in the literature] This study focused on 3D-similarity representative (Q) distribution through finding the best parameter fitted to empirical 13957 x13957 similarity coefficient matrices. For the purpose, EM algorithm and GMM were used. To gain distribution of a query, 1(the query) x13957 similarity coefficients were empirically calculated and fitted to ML estimation. Finally, K-L divergence was calculated from the two distribution. Even though the number of studies using 3D-similarity are enormous, 3D-similarity distribution is rarely mentioned in literature. When comparing our method with the chosen previous studies, we can appeal the three differences. You can also see lines No. 80 to 102 with references [5-20].

1st Difference: other study doesn’t use *distribution* (eg. Torres’ group). Instead of the distribution, network analysis (edge: similarity, node: chemical) such as Torres’ group or machine-learning algorithm based classifiers were used. Almost classifier doesn’t use only chemical similarity but also use other descriptors.

2nd Difference: other study doesn’t generate similarity distribution for *each target* (Medina-Franco’s group, Jain’s group). Instead of the distribution, one distribution of all compounds was visualized or used statistical model without parameter optimization such as Jain’s multinomial Bernoulli model.

3rd Difference: other study doesn’t use *3D-similarity* for the distribution due to uncountable number of conformers (Shoichet’s group, Bajorath’s group). Shoichet’s group built parameterized fitting model but 2D-similarity was used instead of 3D similarity. Bajorath’s group used K-L divergence but their K-L divergence was not used for 3D-similarity distribution.

Reviewer 2 Report

The manuscript is very hard to understand and needs to be largely reworked. Results of the study are not clearly summarized in the abstract; comparison with already available methods is missing; advantages/limitations of the proposed method are not clear; the acronyms and terms are carelessly defined or their definition is missing completely; many mistakes and repetitions.

From what I understand, the main point of the paper is in developing an approach for evaluation similarity/dissimilarity of molecular compounds to/from particular target proteins. For this known inhibitors for each target are clustered using a widely employed Gaussian mixture model (GMM ) algorithms. Then Kullback-Leibler (K-L) divergence is used to determine distances between similarity distributions of compounds withing each a cluster and of those of a query compound to each cluster. Computed K-L distance is then used to asses to which protein target the query compound fits the best and from it is most distinct. This means, that a query compound is practically compared with available dataset of inhibitors for each target, which (i) is a widely-used approach with different techniques readily developed, (ii) does not allow to find any “novel” compounds (or “unprecedented molecular frameworks” as it is stated in the abstract) that would be outside the space of the known inhibitors of the considered target protein. On the other hand, if a query compound is dissimilar to some classes (“unprecedented”), this does not mean that it can be a good inhibitor (i.e. that the result of such comparison has a practical meaning). As far as I see, the only possible novelty of the method is in the way of comparison of a query compound with known inhibitors. In addition, the method accuracy and performance are difficult to judge because (1) the manuscript is very hard to understand and (ii) no comparison with available methods was done and the number of targets used for method evaluation too small to make any statistical assessment.

The major comments:

  1. The statement that compounds are evaluated with respect to “target” proteins is misleading, since in reality, they are compared with different classes of known inhibitors for a particular protein.
  2. Abstract does not show the aim of the study, novelty of the proposed method, and the main results and must be rewritten.
  3. Information in the theoretical background section can be easily found in many textbooks, while the section “Materials and methods” is very shallow. I would suggest to include additionally details on the data preparation/selection that are completely missing in the manuscript: ideas behind the selection of the target proteins for method evaluation, procedure used to select inhibitors for each target; justification why each specific method was chosen (for example, Tanimoto coefficient); what 3D alignment or superposition methods were used, etc.
  4. It is not clear why equations for continuous variable are discussed in the Section “Kullback-Leibler divergence”, while discrete sets are used in the study.
  5. Please include an overview of similar method developed in the field in the Introduction, in particular methods for VS using 3D compound similarity (see, for example, review of Kumar et al Front Chem. 2018;6:315. doi: 10.3389/fchem.2018.00315) and
  6. The aim of the study is developing a method that would provide information on “the most reliable target of the query through transformation of chemical similarity”. Using just four targets for evaluation of such method is clearly not enough to demonstrate method reliability.
  7. Figures: please change figure captions, so that they would be understandable without referring to the main text.
  8. I have not found (maybe I overlooked this…) any discussion on the selection of cluster number (number of Gaussian in GMM model), which has strong effect on the result. The method will not have any practical usefulness unless a reliable way of the cluster number selection is found.
  9. The term “target classes” is misleading, since protein classes are not consideredbut just several proteins are used for method evaluation. Whether they represent different protein classes or not and how they were selected should be discussed in the introduction and the selection criteria should be clearly defined.

The minor comments:

  1. Title: ”compound be compared with drug target classes” – I don’t understand how compounds can be compared with drug targets (i.e. proteins or DNA…); authors probably mean “classes of inhibitors of particular protein target” …
  2. Please explain new or rarely-used in literature terms, such as

(i) “compound-target association” - one can think, for example, that this means just docking…

(ii) “retro-VS”

(iii) “discriminativeness”  - does it mean difference or maybe selectivity?

  1. Abstract:

- “two types of similarity distributions” – this is confusing since GMM is not a similarity distribution per se, this is a method for clustering of data.

 - “While Jaccard-Tanimoto similarity of query-to-ligand pairs could be transformed into query distribution through ML estimation, the similarity of ligand pairs within each target class could be transformed into the representative distribution of a target class through GMM, hyperparameterized through expectation-maximization (EM) algorithm.” -meaning is unclear

- Terms in Equation are not defined

- “Jaccard-Tanimoto similarity of query-to-ligand pairs” - is not explained what is kind of compounds here considered as “query” and what kind as “ligand”. If ligands are those binding to a particular target, the procedure how they were selected should be specified in the manuscript (Method section).

  1. Section Introduction:

-  “The chemical similarity also has contributed to target screening, retro-VS, under the chemocentric assumption, two similar molecules are likely to have similar properties so that they can share biological targets or can show similar pharmacological profile [3-4].”

At the beginning of the sentence target screening (i.e. testing many targets) is mentioned, while at the end “two small molecules.” the authors probably mean compound screening…

- What do author mean by “…to know the most reliable target of the query through transformation of chemical similarity” ?

-“two type similarity distributions: one is from maximum likelihood (ML) estimation of queries

and another is from Gaussian mixture model (GMM) of target classes” – it is not clear (i) what exactly similarity distribution means, (ii)

-“The query-to-ligand similarity tried to be transformed into query distribution through ML estimation.” – meaning is unclear

  1. Figure 1. (a,b) – what do line and arrows mean? (c) “screening of an unprecedented drug scaffold as a retro-virtual screening “- how a novel scaffold can be found if a set of available drug compounds are screened ; “New molecular framework”  - what does it mean?

  1. Section Theoretical backgrounds

- I guess, the phrase “… the probability distributions P(x) andQ(x) replace the Gaussian distributions ..”   has to be changed to  “… the probability distributions P(x) and Q(x) are replaced by….”

- ”Line 123  missing object in  equation contains

  1. Section Results and discussion:

- Please explain exactly how “3D-similarity distributions” is computed.

- “3D similarity matrix, which is occupied by Jaccard-Tanimoto coefficients of (1) query-to-ligand pairs and (2) the ligand pairs within each target class.”  – This does not explain how 3D similarity matrix was built. Is it just two numbers or two 3D similarity matrixes? Usually, 3D similarity assumes a similarity in the shape and physical/chemical properties in the physical space, that seems not be the case…What is the size and meaning of each element of the 3D similarity matrix?

- “made us predict target of the query…. also made us quantify “ “made the query more representative ligand of a class than other queries” – please rephrase

- “heteroatom features (pharmacophore features) using Openeye Toolkit”  - what heteroatom features exactly?

- Line 198 – N was not defined

- Line 203 “epresentative distributions” – representative

- Eq. 3.3: x, delta(x) – are not defined

- Eq.3.4  index i is not defined , why is from 0 to 999?

- Figure 2 It is expected that GMM will provide several distributions, it would be useful to show them separately instead of cumulative plot given. Also it would be useful to demonstrate the difference in  Gaussian components

- Once more, meaning of the term “compound-target association information”  is unclear as it is not directly defined in the text

- “ML estimation”  -not clear what authors mean, according to the reference, this might be the similarity of molecular fingerprints, which does not fit the context

- Line 171-173 sentence repeated.

- Line 181 Openeye Toolkit – reference is missing

- Table 2. Please re-format the images

- Section 4.3 I would suggest to put all data in one table.

- Figure 5.  The colour bars have to be annotated

- Results in sections 4.3.1-4.3.4 can be given in one table

- Line330 “indivisual” – “individual”

- Line 392-394 “a representative of ligands within each target class can be chosen for the comparison” – I would assume from this sentence, that the representative is a ligand, but in the next phrase this representative is defined as a condition (mean of Q-distribution…. outlier of Q-distribution etc.)

  1. Materials and methods

- “under the condition of  ” I would suggest to change to “under the following conditions:”

- “(1) the MMFF94 force field excluding Coulomb interactions & the attractive part of Van der Waals…  deleting hydrogen” – it is not quite clear what forces were taken into account in this case and how reliable conformations can be obtained and comparison can be done if electrostatic is absolutely missing…

-“matrix, python packages such as pandas, numpy, and scipy “ – references are missing

Author Response

To editorial board of International Journal of Molecular Sciences and peer-reviewers,

I, along with my co-authors, revised our manuscript based on the review comments. Now I respond to the comments point by point. Kindly ask you to read the response. For facile reading, we showed marked track-changed (red color) in manuscript and line number of a revised part in this letter.

[Q1-Major comments]

Q1-1. The statement that compounds are evaluated with respect to “target” proteins is misleading, since in reality, they are compared with different classes of known inhibitors for a particular protein.

Ans1-1. Carefully I inform that we didn’t use the statement (that compounds are evaluated with respect to target proteins) in our text. We compared a chemical (query) with a *target class* using 3D-chemocentric approach. One K-L divergence value means ‘the comparison between one compound and one target class. I used the terminology, *target class* to escape the confusion between a target protein and ligands of a targets.

Q1-2. Abstract does not show the aim of the study, novelty of the proposed method, and the main results and must be rewritten.

Ans1-2. Abstract was re-written based on three peer-reviewers’ comments.

[Aim of the study]

quantitative comparison of query compounds to target classes (line No. 32)

[Novelty of the proposed method] (line No. 32-39)

Quantification of the discriminativeness using K-L divergence of 3D similarity distributions

(1) with 3D structures (sampled multi-conformers),

(2) with parameter optimization for the fitting model (EM-GMM, MLE),

and (3) with target annotation (in other words, distribution for respective target class).

[Main results] (line No. 39-45)

(1) Kullback–Leibler (K–L) divergence of each query was calculated and compared between targets.

(2) 3D Similarity-based K–L divergence together with the probability, P(ν(l_m )=i) and the feasibility index, (Fm) showed discriminative power in regard to some query–class associations.

(3) K–L divergence comparison of an unprecedented compound, BNDS-A showed K-L divergence of 3D similarity distributions can be an additional comparison method of known methods to predict the target of a novel compound, such as (1) the rank of 3D similarity score or (2) p-value of one 3D similarity distribution.

Q1-3. Information in the theoretical background section can be easily found in many textbooks, while the section “Materials and methods” is very shallow. I would suggest to include additionally details on the data preparation/selection that are completely missing in the manuscript: ideas behind the selection of the target proteins for method evaluation, procedure used to select inhibitors for each target; justification why each specific method was chosen (for example, Tanimoto coefficient); what 3D alignment or superposition methods were used, etc.

Ans1-3. “Materials and methods section” was revised. Kindly ask you to see the section. In addition, you can see line No. 237-254 in result section.

Q1-4. It is not clear why equations for continuous variable are discussed in the Section “Kullback-Leibler divergence”, while discrete sets are used in the study.

Ans1-4. In this study, we used *fitted distributions* (having continuous variable) for K-L divergence, didn’t directly use *frequent histograms* (having discrete variable) from 3D-similarity matrix.

Q1-5. [similar method] Please include an overview of similar method developed in the field in the Introduction, in particular methods for VS using 3D compound similarity (see, for example, review of Kumar et al Front Chem. 2018;6:315. doi: 10.3389/fchem.2018.00315) and

Ans1-5. Kindly ask you to see line No. 87-102 for overview.Front Chem. 2018;6:315’ is a review article to show how to use 3D similarity. Such article didn’t consider ‘how to compare a group (ligands of each target) and a compound query. The mentioned paper focused on how to describe molecular shape (atom distance based, surface based, and atom-centered Gaussian overlay) and application of shape similarity. Notably, our study focused on distributions from 3D-descripotor and the comparison between the distributions. Focused points are different between our study and the review. Nevertheless, based on the comment, we revised introduction as well as material and methods section.

[Difference between the review and our study]

Front Chem. 2018;6:315

Our study

How to get 3D-similarity score

3 representations * 3 distance metrics

atom-centered Gaussian overlay * Jaccard-Tanimoto coefficient

How to use

Use the ranking of similarity score

Generate distribution of similarity score according to target to compare a query with a target (group)

Q1-6. The aim of the study is developing a method that would provide information on “the most reliable target of the query through transformation of chemical similarity”. Using just four targets for evaluation of such method is clearly not enough to demonstrate method reliability.

Ans1-6. Basically, I agree with the validation on a larger number of targets. This study focused on the methodology for the aim rather than end product of the aim. Due to 3D-conformation & size of each target class, the data point of one target is very more enormous than 2D-similarity distribution [N= No. of compounds* No. of Targets * No. of Conformers]. In particular, similarity matrix made computing burden N^2. Despite the computing burden, we computed the additional four sets within revision time line (April of the special issue) and added addition table (the table3 in this revision).

Q1-7. Figures: please change figure captions, so that they would be understandable without referring to the main text.

Ans1-7. The caption of Figures 1-6 was revised for better readability.

Q1-8. I have not found (maybe I overlooked this…) any discussion on the selection of cluster number (number of Gaussian in GMM model), which has strong effect on the result. The method will not have any practical usefulness unless a reliable way of the cluster number selection is found.

Ans1-8. The K numbers were chosen by hypaerparameter optimization using EM algorithm. Kindly ask you to read line No. 188-215. The effect of K number was described in line No. 385-390 and 441-444.

Q1-9. (1) The term “target classes” is misleading, since protein classes are not considered but just several proteins are used for method evaluation. (2) Whether they represent different protein classes or not and how they were selected should be discussed in the introduction and the selection criteria should be clearly defined.

Ans1-9. (1) In drug discovery, the terminology,target class’, means higher hierarchy such as nuclear receptor, GPCR, kinase, protease, ion channel. We call ‘target class’ as a group of chemical ligands annotated with one target. (2) If we call a target its protein name, I think the calling is more confused with protein, itself. We added definition of the terminology into the list of acronyms. The selection criteria were described in line No. 237-253 of result section.

[Q2-Title]

Q2-1. Title: ”compound be compared with drug target classes” – I don’t understand how compounds can be compared with drug targets (i.e. proteins or DNA…); authors probably mean “classes of inhibitors of particular protein target” …

Ans2-1. The terminology was assigned to name ‘a class of ligands of a protein target’. Brief naming can help readers’ facile reading. We explain the terminology in the list of acronyms.

Q2-2. Please explain new or rarely-used in literature terms, such as

(i) “compound-target association” - one can think, for example, that this means just docking…

Ans. I think that some compound can directly bind to the target (docking) and other compounds can regulate the target indirectly. In addition, some compound regulates a function of the target and another can regulate expression level of target protein. We extracted our data based on single target information but regardless of assay type (eg. cell-level, biochemical level, functional study). That’s the reason why we mentioned “compound-target association”. We add it in the list of acronyms.

(ii) “retro-VS”

Ans. In main text, we defined it at line No. 55, target screening (in other words, retro-VS)

(iii) “discriminativeness” - does it mean difference or maybe selectivity?

Ans. “discriminativeness” is close to difference. I used the terminology based on Kullback and Leibler’s definition. The words,discriminate, discrimination’ exist in Kullback and Leibler’s original article.

[Q3-Abstract]

Q3-1. “two types of similarity distributions” – this is confusing since GMM is not a similarity distribution per se, this is a method for clustering of data.

Ans3-1. Gaussian mixture model (GMM) was used to generate representative distribution of a target class. In detail, hyper-parameters of GMM was used to get the best distribution fitted to frequent histograms from 3D- similarity matrix.

Q3-2. “While Jaccard-Tanimoto similarity of query-to-ligand pairs could be transformed into query distribution through ML estimation, the similarity of ligand pairs within each target class could be transformed into the representative distribution of a target class through GMM, hyperparameterized through expectation-maximization (EM) algorithm.” -meaning is unclear

Ans3-2. In order to improve the readability, I redrew Figure 1 to describe abstract and introduction. In addition, two times English-editing applied to the whole part of the manuscript. In addition, we explain ‘Jaccard-Tanimoto similarity of query-to-ligand pairs’ in the list of acronyms.

Q3-3. Terms in Equation are not defined

- “Jaccard-Tanimoto similarity of query-to-ligand pairs” - is not explained what is kind of compounds here considered as “query” and what kind as “ligand”. If ligands are those binding to a particular target, the procedure how they were selected should be specified in the manuscript (Method section).

Ans3-3. [Terms in Equation] You can see the list of acronyms and line No. 258 to 265 as well as line No. 296 to 303. [Jaccard-Tanimoto similarity of query-to-ligand pairs] For the first time, I kindly request you to read *Fig1(d)* for the understanding. “ligand” is a member of a target class and “query” can be any compound regardless of target class for the comparison. In this study, every sampled compound of 4 target classes (13,957 conformers x 4 target classes) was used as a query to show the performance of K-L divergence. In addition, BNDS-A compound is only one query not exiting in any target class. Kindly see line No. 539-543.

[Q4-Introduction]

Q4-1. “The chemical similarity also has contributed to target screening, retro-VS, under the chemocentric assumption, two similar molecules are likely to have similar properties so that they can share biological targets or can show similar pharmacological profile [3-4].” At the beginning of the sentence target screening (i.e. testing many targets) is mentioned, while at the end “two small molecules.” the authors probably mean compound screening…

Ans4-1. For clear understanding, I separate the sentence to two sentences: The chemical similarity also has contributed to target screening, retro-VS, under the chemocentric assumption. Chemocentric assumption means two similar molecules are likely to have similar properties so that they can share biological targets or can show similar pharmacological profile.

Q4-2. What do author mean by “…to know the most reliable target of the query through transformation of chemical similarity” ?

Ans4-2. I revised the part: the most reliable target -> primary target

Q4-3. “two type similarity distributions: one is from maximum likelihood (ML) estimation of queries and another is from Gaussian mixture model (GMM) of target classes” – it is not clear (i) what exactly similarity distribution means,

Ans4-3. For the first time, kindly remind you the Figure 1(d).

1) 13957 x 13957 similarity matrix of a target class -> frequent histogram (bin number =1000) -> find the best GMM hyper-parameters fitted to the histogram -> representative (Q) similarity distribution from the parameters

2) 13957 x 1 similarity coefficient between a query and ligands of a target class -> frequent histogram (bin number =100) -> find the best MLE parameters fitted to the histogram -> similarity distribution from the parameters

In addition, kindly ask you to read revised Figure1 and Ans3-1. The sentence was refined: line No. 103-105.

Q4-4. (ii) “The query-to-ligand similarity tried to be transformed into query distribution through ML estimation.” – meaning is unclear

Ans4-4. Kindly ask you to read revised Figure1 and Ans4-3. In addition, the sentence was refined: line No. 229-230.

Q4-5. Figure 1. (a,b) – what do line and arrows mean? (c) “screening of an unprecedented drug scaffold as a retro-virtual screening “

Ans4-5. We redrew the Figure1. The arrow of Fig(a,b) means comparison.

Q4-6. how a novel scaffold can be found if a set of available drug compounds are screened ; “New molecular framework” - what does it mean?

Ans4-6. If you have a novel scaffold dissimilar to known bioactive compounds, you can predict probable target using K-L divergence between the scaffold (query) and target classes. New molecular framework means a new Bemis-Murcho framework. It is similar to scaffold.

Prior

Posterior

Question

A set of available drug compounds

Find a novel scaffold

This Study

Having a novel scaffold

Predicting the profiles of the scaffold

[Q5- Theoretical backgrounds]

I guess, the phrase “… the probability distributions P(x) and Q(x) replace the Gaussian distributions ..” has to be changed to “… the probability distributions P(x) and Q(x) are replaced by….” ”Line 123 missing object in equation contains

Ans5. English editing retain the expression. Line No. 150-151.

We revised the part (previous Line 123). Line No. 157-162.

[Q6- Results and discussion]

Q6-1. Please explain exactly how “3D-similarity distributions” is computed.

Ans 6-1. The practical process was added into Materials and methods exactly.

Step1. 3D- similarity matrix (line No. 258-265, 533-543)

Step2. Binning of 3D- similarity coefficient from Step1 matrix (line No. 268-275)

Step3. Fitting of the binned data (step2) into GMM or MLE == find hyperparameter of GMM (line No. 189-215) or MLE (https://github.com/college-of-pharmacy-gachon-university/KLD-Pharmacological_Class_Similarity)

In addition, you can see Figure 1(d).

Q6-2. “3D similarity matrix, which is occupied by Jaccard-Tanimoto coefficients of (1) query-to-ligand pairs and (2) the ligand pairs within each target class.” – This does not explain how 3D similarity matrix was built. Is it just two numbers or two 3D similarity matrixes? Usually, 3D similarity assumes a similarity in the shape and physical/chemical properties in the physical space, that seems not be the case…What is the size and meaning of each element of the 3D similarity matrix?

Ans 6-2. [How to build 3D similarity matrix] Using shape toolkit of Openeye, we made python script to build the similarity matrix (reading 3D-structure of every pair -> alignment -> calculate similarity score -> sorting scores -> writing the scores into the matrix). Even though the code is similar to our previous publication (supplementary code: conformer_selection.py, J. Cheminformatics, 2017, 9, 21) as well as cookbook of Openeye toolkit, we add our refined code into supplementary information (https://docs.eyesopen.com/toolkits/python/shapetk).

In detail, the constructors were used as follows:

OEOverlay() : optimization of the alignment(overlap) between query and database

OEBestOverlayScoreIter(): sorting all scores to highest tanimoto coefficient before writing similarity score into an empty matrix.

[Is it just two numbers or two 3D similarity matrixes?]

The number of matrixes for (1) query-to-ligand pairs: No. (target)* No. (query)

The dimension of a matrix for (1) query-to-ligand pairs: 1* 13957

The number of matrixes for (2) the ligand pairs within each target class: No. (target)

The dimension of a matrix for (2) the ligand pairs within each target class: 13957 *13957

[each element of the 3D similarity matrix] one element of a 3D similarity matrix is 3D-similarity score.

Q6-3. “made us predict target of the query…. also made us quantify “ “made the query more representative ligand of a class than other queries” – please rephrase

Ans 6-3. We revised the part. Kindly see line No. 231-236.

Q6-4. “heteroatom features (pharmacophore features) using Openeye Toolkit” - what heteroatom features exactly?

Ans 6-4. In general, a heteroatom in cheminformatics and chemistry means an atom except for hydrogen and carbon. Therefore, ‘heteroatom feature’ means position (Cartesian coordinate) of a heteroatom in a query.

Q6-5. Line 198 – N was not defined

Ans 6-5. N at line198 means ‘dimensions’.

Q6-6. Line 203 “epresentative distributions” – representative

Ans 6-6. It was revised.

Q6-7. Eq. 3.3: x, delta(x) – are not defined

Ans 6-7. We defined them (line No. 272-273).

x = 3D-similarity score (Jaccard-Tanimoto coefficient)

the range of x is [0,2]

delta(x) = dx_{i+1} - dx_i = (range of x)/(total number of bins) = 2/(999+1), i = 0 to 999 (eq. 3.4)

Q6-8. Eq.3.4 index i is not defined , why is from 0 to 999?

Ans 6-8. ‘i’ is index for the order of bin, [x_i,x_{i+1}], of continuous similarity scores. Since the probability density (3.4) results from the scores of a 3D-simialrity matrix, when number of bins is large enough, the density reflect property of the scores. Therefore, although the range of similarity scores, [0,2] is small, range of index, [0,999] is relatively large.

Q6-9. Figure 2 It is expected that GMM will provide several distributions, it would be useful to show them separately instead of cumulative plot given. Also it would be useful to demonstrate the difference in Gaussian components

Ans 6-9. The figure 2 well shows (1) the difference between Gaussian components and (2) fitting with raw data (frequent histogram). We put 2(a)-(d) separately.

Q6-10. Once more, meaning of the term “compound-target association information” is unclear as it is not directly defined in the text

Ans 6-10. “compound-target association information”. We add it to terminology list. In addition, kindly ask you to see the answer on Q2-2.

Q6-11. “ML estimation” -not clear what authors mean, according to the reference, this might be the similarity of molecular fingerprints, which does not fit the context

Ans 6-11. ML estimation means Maximal Likelihood estimation, which method of estimating the parameters of a probability distribution by maximizing a likelihood function, so that under the assumed statistical model the observed data is most probable. In this study, since the statistical model is normal distribution, the parameters are mean mu and standard deviation, sigma, as shown in (3.10), when sigma=sigma_1 and mu= mu_1, likelihood function is maximized and g(x: mu_1, sigma_1) is the most similar distribution with the distribution from observed data. We add the terminology into the list of acronyms.

Q6-12. Line 171-173 sentence repeated.- Line 181 Openeye Toolkit – reference is missing

Ans 6-12. We revised the repetition. We added the reference.

Q6-13. Table 2. Please re-format the images

Ans 6-13. We refined the table 2. In addition, 2D-structure of queries were moved to supplementary information (Table3).

Q6-14. Section 4.3 I would suggest to put all data in one table.

Ans 6-14. If your suggestion means Table2 and Table 4, they cannot be combined due to different selection criteria. Queries of Table2 were manually chosen based on structure uniqueness and Table4 showed representative queries based on Q distribution and unprecedented scaffold, BNDS-A.

Q6-15. Figure 5. The colour bars have to be annotated

Ans 6-15. They were annotated in our revision.

Q6-16. Results in sections 4.3.1-4.3.4 can be given in one table

Ans 6-16. They are already existing in Table2 (first ligand of ESR). They were described to help readers’ understanding our result and Table2.

Q6-17. Line330 “indivisual” – “individual”

Ans 6-17. We revised the wrong word.

Q6-18. Line 392-394 “a representative of ligands within each target class can be chosen for the comparison” I would assume from this sentence, that the representative is a ligand, but in the next phrase this representative is defined as a condition (mean of Q-distribution…. outlier of Q-distribution etc.)

Ans 6-18. The sentence was revised. Line No. 453-458.

Here is the process to get the representative ligands

3D-sim matrix of ligands -> histogram -> Q-distribution, Q(x), x= 3D-similarity score -> Get the values: Mean, (Mean + 2SD), (Mean - 2SD) of 3D-similarity score -> Go to the initial 3D-sim matrix -> Find the representative ligands having the nearest score to Mean, (Mean + 2SD), and (Mean - 2SD).

[Q7- Materials and methods]

Q7-1. “under the condition of ” I would suggest to change to “under the following conditions:”

Ans 7-1. We revised it.

Q7-2. “(1) the MMFF94 force field excluding Coulomb interactions & the attractive part of Van der Waals… deleting hydrogen” – it is not quite clear what forces were taken into account in this case and how reliable conformations can be obtained and comparison can be done if electrostatic is absolutely missing…

Ans 7-2. We revised the part. Kindly see line No. 491-499.

Q7-3. “matrix, python packages such as pandas, numpy, and scipy “ – references are missing

Ans 7-3. We added the references [43-47]

Reviewer 3 Report

A comprehensive article presented from the authors. However, the authors may consider the followings:

  1. Please elaborate (or give rationale) why in particular, these target classes were chosen, “estrogen receptor alpha (ESR), vitamin D 26 receptor (VDR), cyclooxygenase-2 (COX2), and cathepsin D (CTSD)”
  2. Please state clearly the novelty of this research in your abstract and conclusion.
  3. Please also correlate any alignments of the current work to that of Shin et al 2015“Shin et al 2015 PMID: 26193243”.
  4. Figure 1a to 1c should be revised for a clearer display of the theme.
  5. The author should be consistent in their font size and format.
  6. In Figure 2, please improve the figure resolution.
  7. The authors should check for their use of English by professionals.
  8. In Table 1, please improve the presentation of the table.
  9. As for Figure 3 to 6, the author may consider organizing the spacing in putting the Figure subtitles to identity its component clearly

Author Response

To editorial board of International Journal of Molecular Sciences and peer-reviewers,

I, along with my co-authors, revised our manuscript based on the review comments. Now I respond to the comments point by point. Kindly ask you to read the response. For facile reading, we showed marked track-changed (red color) in manuscript and line number of a revised part in this letter.

Peer-reviewer 2

[Q1-Rationale] Please elaborate (or give rationale) why in particular, these target classes were chosen, “estrogen receptor alpha (ESR), vitamin D 26 receptor (VDR), cyclooxygenase-2 (COX2), and cathepsin D (CTSD)”
Ans1. Due to long full name, we used UniProt abbreviation.

[Q2-Novelty] Please state clearly the novelty of this research in your abstract and conclusion.

Ans2. Our described novelty was re-written for impressive reading in main text and compared with literature. Based on the revision, abstract (line No. 32-39) and conclusion (line No. 550-565) were revised to show the novelty of this study.

[Q3-Figures]

Ans3-1. Figure 1a to 1c were revised for a clearer display of the theme.

Ans3-2. For improved figure resolution, Figure 2 were separately uploaded.

Ans3-3. As for Figure 3 to 6, the spacing of Figure subtitles were reorganized to identity its component clearly.

Ans3-4. You mentioned Table 1 but I guessed your intention wad figure of Table2. The figures of Table2 were moved to supplementary information (Table3).

Ans3-5. Font & size were adjusted.

[Q4-Method & Reference] Please also correlate any alignments of the current work to that of Shin et al 2015“Shin et al 2015 PMID: 26193243”.

Ans3-6. The mentioned paper focused on 3D-descriptor and our study focused on distributions from 3D-descripotor and the comparison between the distributions. Focused points are different between our study and PMID: 26193243. However, based on the comment, we revised our material and methods section. In particular, alignment algorithm was concretely described in the section. In addition, you can see lines No. 80 to 102 with references [5-20] including Shin’s article.

Round 2

Reviewer 1 Report

I appreciate the authors' efforts in improving the manuscript. The overall presentation (including figures and tables) is much clearer now. The supplementary information provide sufficient background.